# The central histaminergic system slows visual processing in the retina and lateral geniculate nucleus of awake mice

Matteo Tripodi, Hiroki Asari *

Epigenetics and Neurobiology Unit, EMBL Rome, European Molecular Biology Laboratory, Monterotondo, Italy

* asari@embl.it

## Abstract

Sensory processing is dynamically modulated by an animal's behavior and internal states. Growing evidence suggests that such modulation starts from early stages, already at the retina, but the underlying mechanisms remain elusive. Combining pharmacological and chemogenetic tools with single-unit extracellular recordings in awake head-fixed mice, here we identified that the visual responses of retinal ganglion cells and the lateral geniculate nucleus were both made weaker and slower by histaminergic projections from the tuberomammillary nucleus of the posterior hypothalamus. The observed changes in the visual responses were, however, not directly linked with histaminergic modulation of pupil dynamics or locomotion behavior. Instead, our computational modeling analysis suggests that the temporal response modulations arise from changes in the intrinsic properties of the circuit, such as gain modulation via the histamine H1 receptors in the retina. Facilitation of visual processing at low histamine levels may offer an ethological advantage, enabling animals to respond rapidly to visual threats during periods of reduced activity.

## Introduction

In the mammalian brain, histaminergic neurons are exclusively located in the tubero-mammillary nucleus (TMN) and surrounding areas of the posterior hypothalamus [1]. They express histidine decarboxylase (HDC) and innervate widely across the brain [2], including the retina [3–6]. These neurons remain silent during sleep, but increase their firing during wakefulness and reach their maximal activity at the state of high vigilance, much as other monoaminergic systems do, such as the serotonergic neurons in the dorsal raphe nucleus (DRN) of the brain stem [2,7,8]. Brain histamine exerts its effects on neurons via three types of receptors: the H1 and H2 receptors mediate excitation of postsynaptic cells, while the H3 receptors serve as an autoreceptor to presynaptically inhibit histamine release [1]. The role of the H4 receptors in the central nervous system remains unknown [9]. The central histaminergic system

**Data availability statement:** All relevant data and codes are available from ZENODO (https://doi.org/10.5281/zenodo.17016431).

**Funding:** This work was supported by research grants from EMBL (H.A.). The funder had no role in study design, data collection and analysis, decision to publish, or preparation of the manuscript.

**Competing interests:** The authors have declared that no competing interests exist.

**Abbreviations:** AAVs, adeno-associated viruses; AP, Anterior–Posterior; DRN, dorsal raphe nucleus; DS, direction-selective; DSI, DS index; DV, Dorsal–Ventral; FPS, frames per second; GABA, γ-aminobutyric acid; HDC, histidine decarboxylase; IQR, interquartile range; LEDs, light-emitting diodes; LGN, lateral geniculate nucleus; ML, Medial–Lateral; OS, orientation-selective; OT, optic tract; PFA, paraformaldehyde; PSAM, pharmacologically selective actuator module; PSEM, pharmacologically selective effector molecule; RF, receptive field; RGCs, retinal ganglion cells; SD, standard deviation; STA, spike-triggered average; TMN, tuberomammillary nucleus; UV, ultra-violet.

has been implicated in regulating an animal's internal state and a broad range of behavior, including the sleep–wake cycle [10,11], arousal [12,13], suppression of food intake [14–16], and learning [17–20]. Histaminergic neurons, however, corelease γ-aminobutyric acid (GABA) [21] and interact with other neuromodulatory systems, such as the cholinergic and serotonergic systems [2,22,23]. This makes it difficult to dissect the effects of histamine by itself on neural circuit functions. In particular, histaminergic modulation of sensory processing remains largely elusive [24,25].

The retina performs the first stage of visual processing, and conveys visual information to the brain via spike trains of retinal ganglion cells (RGCs). Growing evidence supports that the activity of RGCs is correlated with various behavioral measures, such as locomotion and pupil size [26–29], pointing to a more dynamic role of the retina than is assumed for a deterministic processor. The vertebrate retina is part of the central nervous system equipped with both intrinsic and extrinsic neuromodulation mechanisms, much as other brain regions are [24,25]. For instance, dopamine is synthesized and released by a group of retinal amacrine cells under the control of circadian rhythm, and contributes to adjusting the retinal circuitry for rod or cone vision by modulating gap-junction couplings [30]. Such retinal intrinsic mechanisms of neuromodulation are, however, much slower than needed to modulate retinal function to follow rapid behavioral changes [31,32]. To this end, the retina seems to exploit the centrifugal system to receive inputs from a variety of brain regions in a species-specific manner [4]. For example, histaminergic and serotonergic projections from TMN and DRN, respectively, have been anatomically described in the mammalian retina [3,5,6,33]. However, compared to the isthmo-optic pathway in birds [34] or the olfacto-retinal circuit in fish [35], little is known about the function of the mammalian centrifugal system [36,37].

Combining in vivo single-unit extracellular recordings from either the lateral geniculate nucleus (LGN) or the optic tract (OT) with pharmacological and chemogenetic tools, here we explored how histamine affects the early visual processing in awake head-fixed mice. During the recording sessions, we simultaneously monitored the subject animal's behavior, such as pupil dynamics and locomotion, to better correlate the effects of histamine on the visual response properties with those on the behavioral measures. We also took a computational approach to clarify the net effect of histamine on the visual responses from functional viewpoints. Because visual processing in the brain fully relies on the signals from the retina, characterization of the retinal operation in vivo is indispensable to better understand how the visual system works.

## Results

### Histamine leads to weaker and slower visual responses in the mouse lateral geniculate nucleus

To investigate how histamine influences the mouse visual system, we first performed in vivo single-unit extracellular recordings from the LGN in awake head-fixed mice (Figs 1A, 1B, and S1), and compared the visual response properties of the recorded

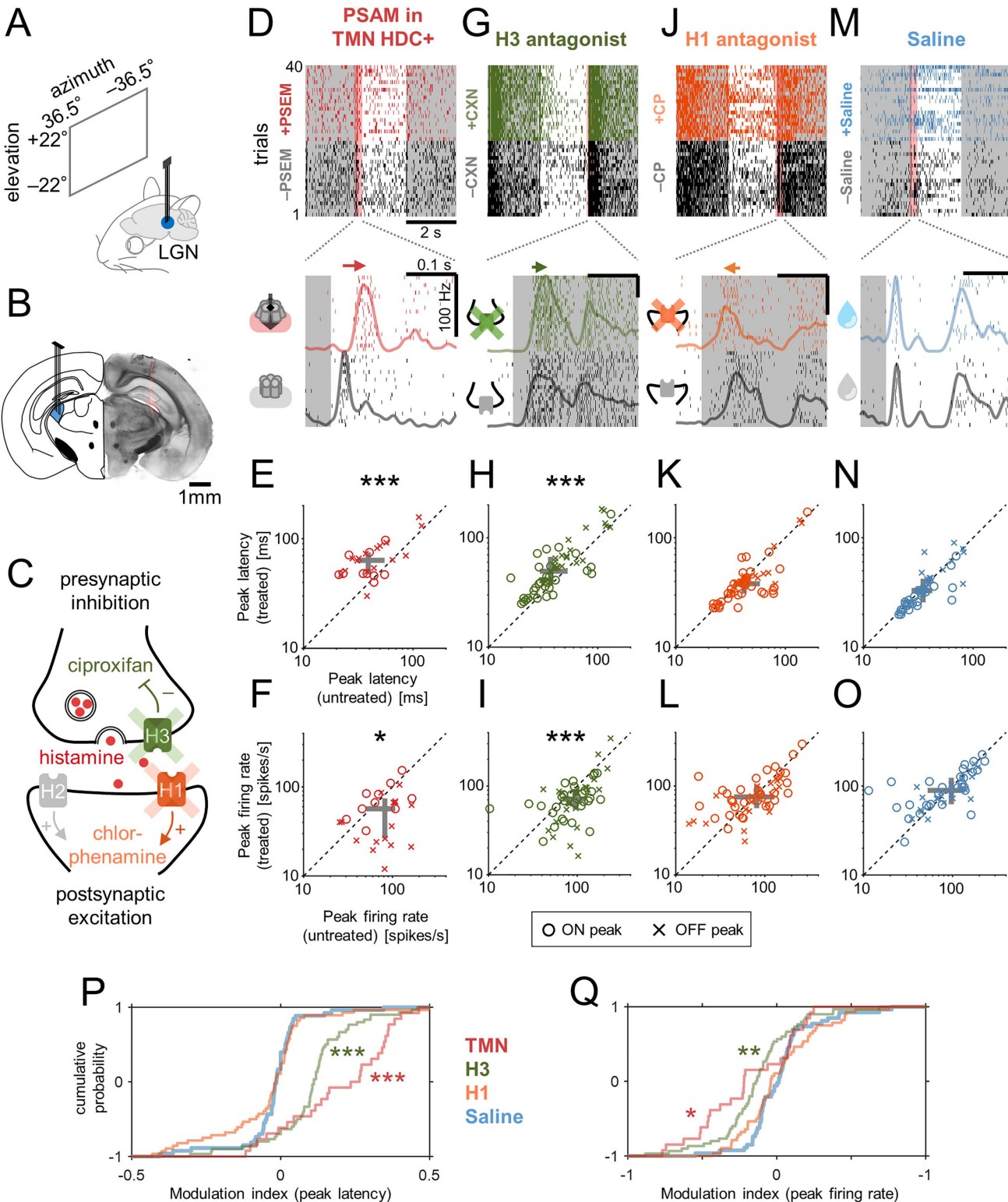

**Fig 1. Histamine reduced LGN visual responses in awake head-fixed mice. (A,B)** Schematic diagram of LGN recordings (A) and a representative histological image showing the electrode location (B; DiI stain, red). **(C)** Schematic diagram of histamine receptor localizations and their cellular function. Chlorphenamine blocks the postsynaptic receptor H1 (orange) that mediates the action of histamine postsynaptically, while ciproxifan blocks

the autoreceptor H3 (green) that inhibits presynaptic histamine release. **(D)** Visual responses of a representative LGN cell to full-field contrast-inverting stimuli (2 s intervals) before and after chemogenetically activating HDC+ cells in TMN: *top*, spike raster across trials; *bottom*, zoom-in of the spike raster around stimulus onset (−50 to 250 ms, red shade on top) and peri-stimulus time histogram, showing a delay of the onset response after the chemogenetic treatment. **(E,F)** Pairwise comparison of the LGN population responses before and after the chemogenetic treatment (E, peak latency: untreated, $39 \pm 17$ ms, median ± median absolute deviation; treated, $64 \pm 20$ ms; $p < 0.001$, Wilcoxon signed-rank test with Bonferroni correction; F, peak firing rate: untreated, $82 \pm 33$ Hz; treated, $57 \pm 26$ Hz; $p = 0.04$, $n = 26$ LGN cells from 3 animals): circle, ON peak; cross, OFF peak; gray, inter-quartile range. **(G–I)** Corresponding data before and after ciproxifan administration (G, representative cell's response, with a zoom-in around stimulus offset; H, peak latency: $37 \pm 22$ ms vs. $50 \pm 26$ ms, $p < 0.001$; I, peak firing rate: $95 \pm 35$ Hz vs. $71 \pm 31$; $p < 0.001$, $n = 60$ LGN cells from 4 animals). **(J–L)** Corresponding data before and after chlorphenamine administration (J, representative cell's response; K, peak latency: $42 \pm 20$ ms vs. $38 \pm 16$ ms, $p = 0.3$; L, peak firing rate: $81 \pm 43$ Hz vs. $76 \pm 33$; $p = 0.5$, $n = 56$ LGN cells from 3 animals). **(M–O)** Corresponding data before and after saline administration (M, representative cell's response; N, peak latency: $35 \pm 22$ ms vs. $33 \pm 21$ ms, $p = 0.2$; O, peak firing rate: $97 \pm 48$ Hz vs. $89 \pm 34$; $p = 0.5$, $n = 53$ LGN cells from 3 animals). **(P,Q)** Cumulative distributions of modulation indices before and after each treatment (in corresponding colors): P, peak latencies; Q, peak firing rate; * $p < 0.05$; ** $p < 0.01$; *** $p < 0.001$ from the post-hoc test against the saline (control) condition on the average group ranks (Kruskal–Wallis test). Data and code underlying this figure are available at https://doi.org/10.5281/zenodo.17016431.

cells before and after perturbing the central histaminergic system (Figs 1 and 2; see Materials and methods for details). Besides directly examining the responses to repeatedly presented visual stimuli, we employed stimulus ensemble statistical techniques ("reverse correlation") to systematically characterize the visual response properties of the recorded cells in the framework of a linear-nonlinear cascade model [29]. In particular, using full-field white-noise stimuli, we analyzed histaminergic effects on (1) the linear temporal filter, estimated by the spike-triggered average (STA) stimulus—i.e., the mean stimulus that triggered spiking responses—under a given behavioral state; and (2) the static nonlinear gain function, i.e., an instantaneous mapping of the STA output to the neural responses (Fig 2B). We also functionally mapped the receptive field (RF) of the cells by calculating the linear spatial filter from their responses to random binary "checkerboard" stimuli, where the contrast of each spatial location was randomly and independently inverted over time. To manipulate the histaminergic system, we used two different approaches (S2 Fig). The first was the chemogenetic activation of the central histaminergic system [38]. We injected Cre-dependent adeno-associated viruses (AAVs) carrying pharmacologically selective actuator module (PSAM)-5HT3HC channels into the TMN of the posterior hypothalamus in HDC-Cre transgenic mice, and then achieved selective activation of the target neurons by systemic administration of the pharmacologically selective effector molecule (PSEM; 5 mg/kg). The second approach was the use of pharmacological tools in wild-type mice (Fig 1C) [8]. Specifically, we used chlorphenamine (5 mg/kg) [39,40], an antagonist of a postsynaptic histamine receptor H1; and ciproxifan (12 mg/kg) [41,42], an antagonist of the H3 auto-receptor that presynaptically inhibits histamine release. These drugs effectively down- and up-regulate the action of histamine in the nervous system, respectively. Thus, resulting changes in visual processing, if any, were expected to be in opposite directions. While we let the animals move freely under the head-fixed condition, the reverse correlation analysis was done on the periods when the animals stayed stationary (<2 cm/s) to minimize the side effect of locomotion behavior (S3 Fig) [26,43].

Upon chemogenetically activating HDC+ neurons in TMN, we found that LGN neurons generally showed weaker and slower visual responses. The peak responses to stimulus onset and offset were both significantly reduced in amplitude and prolonged in latency (Fig 1D–1F). Moreover, the mean firing rate during the white-noise visual stimulation became significantly lower (median, −4.7 Hz; interquartile range (IQR), 7.9 Hz; $p < 0.001$, Wilcoxon signed-rank test here and thereafter unless otherwise noted; $n = 52$ from 3 animals); and the response gain also significantly decreased (median, −33%; IQR, 55%; $p < 0.001$) that we computed as the ratio of the estimated static nonlinearity functions before and after PSEM injection (Fig 2C and 2G). The temporal filter became elongated regardless of their functional cell types (S4 Fig), leading to significantly longer peak latencies (median, +31 ms; IQR, 33 ms; $p < 0.001$) and lower peak frequencies (median, −16%; IQR, 47%; $p < 0.001$).

Consistent results were obtained when histamine release was pharmacologically up-regulated. Specifically, after the application of ciproxifan (H3 antagonist), both the onset and offset peak responses became weaker and longer (Fig 1G–1I). Furthermore, during white-noise stimulation, we found significantly reduced mean firing rates (median, −3.0 Hz;

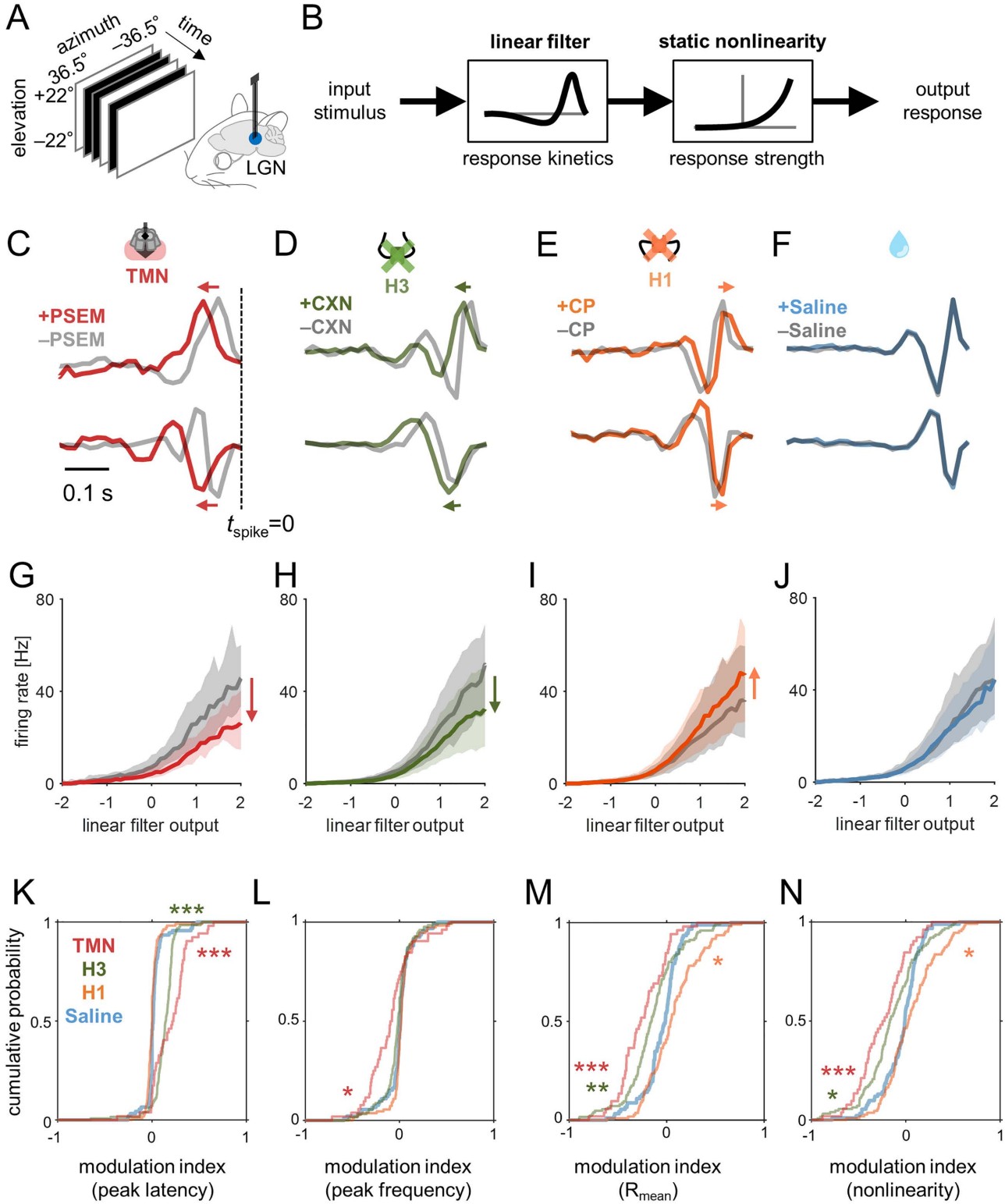

**Fig 2. Histamine made the visual responses of LGN weaker and slower in awake head-fixed mice. (A,B)** Schematic diagram of LGN recordings (A) and linear-nonlinear cascade model **(B)**. Reverse-correlation was used to estimate the linear filter and static nonlinearity that represent response kinetics and strength, respectively. **(C)** Linear filters of two representative LGN cells before (gray) and after chemogenetic activation of HDC+ cells in

TMN (red). We focused only on those data when the animal stayed stationary (<2 cm/s) to minimize the side effect of locomotion behavior (S3 Fig). **(D–F)** Representative LGN linear filters before (gray) and after pharmacological treatment (D, 12 mg/kg ciproxifan, green; E, 5 mg/kg chlorphenamine, orange) or saline injection (F). **(G–J)** Static nonlinearities averaged across LGN population before and after treatment in corresponding colors (G, chemogenetics, $n = 52$ from 3 animals; H, ciproxifan, $n = 126$ from 4 animals; I, chlorphenamine, $n = 110$ from 3 animals; J, saline, $n = 92$ from 3 animals): line, median; gray shade, 25 and 75 percentiles. **(K–N)** Cumulative distributions of the modulation index before and after each treatment (in corresponding color): peak latency (K) and frequency (L) of the linear temporal filters, mean evoked firing rate during stimulation (M), and static nonlinearity (i.e., response gain; N): * $p < 0.05$; ** $p < 0.01$; *** $p < 0.001$ from the post-hoc test against the saline control on the average group ranks (Kruskal–Wallis test). Data and code underlying this figure are available at https://doi.org/10.5281/zenodo.17016431.

IQR, 5.5 Hz; $p < 0.001$; $n = 126$ cells from 4 animals) and response gain (median, −23%; IQR, 65%; $p = 0.002$); and the linear temporal filters showed significantly longer peak latency (median, +16 ms; IQR, 11 ms; $p < 0.001$) and lower peak frequency (median, −39%; IQR, 20%; $p = 0.08$; Fig 2D and 2H). These effects of the pharmacological treatment were less prominent than those of the chemogenetic manipulation (Figs 1P, 1Q, and 2K–2N). None of these treatments significantly affected the estimated RF size at the population level (S5 Fig).

In contrast, chlorphenamine (H1 antagonist) led to marginal changes in the visual response properties of LGN neurons in the opposite direction. The onset/offset stimulus responses were faster in some cells, but the changes were not statistically significant at the population level (Fig 1J–1L). Likewise, we observed a tendency towards an increased mean evoked firing rate (median, +1.1 Hz; IQR, 6.8 Hz; $p = 0.09$; $n = 110$ cells from 3 animals) and response gain (median, +14%; IQR, 94%; $p = 0.015$), and facilitated response kinetics with significantly higher peak frequency of the linear temporal filter (median, +4%; IQR, 15%; $p = 0.02$) but no significant changes in its peak latency (median, 0 ms; IQR, 5 ms; $p = 0.7$) or the RF size (S5 Fig). Taken together, these results indicate that histamine reduces the response gain and slows down the response kinetics in the mouse LGN.

## Histaminergic modulation starts from the mouse retina

Histaminergic neurons are located solely in the hypothalamus, most densely in TMN, and project their axons widely across the brain [2], including the mouse retina [3,4,6]. To test if the observed histaminergic modulation of the visual processing in LGN originates in the retina, we next performed in vivo OT recordings to monitor the retinal output responses in awake head-fixed mice (Fig 3A and 3B) [29] before and after manipulating the central histaminergic system (Figs 3 and S6–S9).

Consistent with the results in LGN, chemogenetic activation of HDC+ neurons in TMN led to reduced firing of RGCs (median, −9.8 Hz; IQR, 13.1 Hz; $p = 0.01$; $n = 26$ from 4 animals) with lower response gain (median, −28%; IQR, 57%; $p = 0.01$); and slower response kinetics with elongated linear temporal filters (Fig 3C and 3G; peak latency change, median, 10 ms; IQR, 33 ms; $p < 0.001$; peak frequency change, median, −17%; IQR, 34%, $p = 0.003$). We also found that the latencies of the peak responses to stimulus onset and offset were both significantly longer, with a reduced peak firing (S6A and S6F Fig). In contrast, no change in these response parameters was observed when targeting HDC+ populations in the anterior hypothalamus outside TMN for the PSAM/PSEM activation (firing rate change, median, 1.7 Hz, IQR, 16.5 Hz, $p = 0.6$; gain change, median, 8%, IQR, 99%, $p = 0.5$; latency change, median 0 ms, IQR, 12 ms, $p = 0.6$; frequency change, median −6%, IQR, 17%, $p = 0.09$; $n = 23$ from 3 animals). Neither the onset nor the offset stimulus response was affected (S6B and S6G Fig). Anterograde tracing of HDC+ cells—via injection of Cre-dependent AAVs encoding fluorescent marker proteins into the TMN of HDC-Cre animals—showed labeled axons in the optic chiasm and/or the optic nerve (3 out of 9 animals; S2C and S2D Fig). Unlike previous reports [3,4,6], however, we did not identify any labeled axons in the retinal tissues (e.g., S2E Fig), likely because these projections are sparse and located far from the cell bodies, making the signals too weak. Nonetheless, our data collectively support that the histaminergic modulation of the early visual system is mediated by the centrifugal projections specifically from TMN (Figs 3G–3J and S2).

The effects of the pharmacological treatments were overall similar between LGN and RGCs, though certain differences existed. First, pharmacologically blocking the H3 receptors with ciproxifan led to slower response kinetics in both LGN and

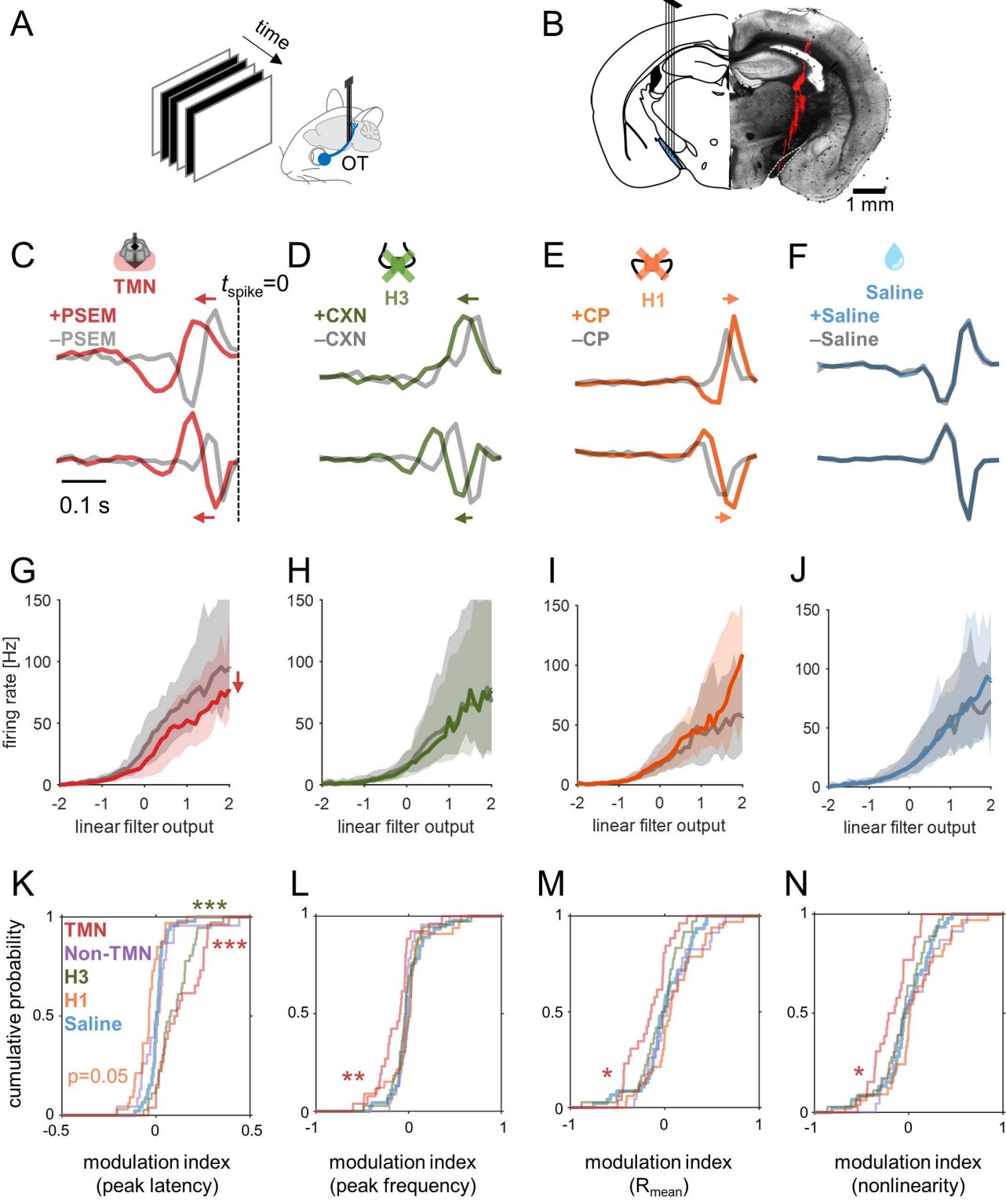

**Fig 3. Histamine primarily affected the visual response kinetics of RGCs. (A,B)** Schematic diagram of in vivo optic tract (OT) recordings (A) and a representative histological image showing the electrode location (B; DiI stain, red). **(C–F)** Linear filters of two representative RGCs before (gray) and after chemogenetic (A, red) or pharmacological treatment (B, 12 mg/kg ciproxifan, green; C, 5 mg/kg chlorphenamine, orange; D, saline, blue). **(G–J)**

Static nonlinearities averaged across RGC population before and after treatment in corresponding colors (G, chemogenetics, $n = 26$ cells from 4 animals; H, ciproxifan, $n = 36$ from 4 animals; I, chlorphenamine, $n = 33$ from 3 animals; J, saline, $n = 46$ cells): line, median; gray shade, 25 and 75 percentiles. **(K–N)** Cumulative distributions of the modulation index before and after each treatment (in corresponding color; $n = 23$ from 3 animals for chemogenetic treatment of non-TMN HDC+ cells): peak latency (K) and frequency (L) of the linear temporal filters, mean evoked firing rate during stimulation (M), and static nonlinearity gain function (N); * $p < 0.05$; ** $p < 0.01$; *** $p < 0.001$ from the post-hoc test against the saline control condition in the average group ranks (Kruskal–Wallis test). Data and code underlying this figure are available at https://doi.org/10.5281/zenodo.17016431.

RGCs (latency change, median, 10 ms; IQR, 14 ms; $p < 0.001$; frequency change, median, −7%; IQR, 25%; $p = 0.12$; $n = 36$ from 4 animals), but the firing rates and the response gain remained largely unaffected in RGCs (Fig 3D and 3H; firing rate change, median −0.1 Hz; IQR, 12.7 Hz; $p = 0.3$; gain change, median, −7%; IQR, 69%, $p = 0.5$). This was also the case with the peak onset/offset stimulus responses (S6C and S6H Fig). Second, while chlorphenamine (H1 antagonist) drove faster kinetics with marginally higher firing in both LGN and RGCs, the effects on the temporal filters were primarily on the peak frequencies in LGN (Fig 2E) but on the peak latencies in RGCs (Fig 3E; latency change, median, −5 ms; IQR, 8 ms; $p = 0.006$; frequency change, median, +4%; IQR, 25%; $p = 0.6$; firing rate change, median, +2 Hz, IQR, 11 Hz; $p = 0.09$; gain change, median, 4%, IQR, 90%, $p = 0.3$; $n = 33$ RGCs from 3 animals). Likewise, a shorter peak latency was observed in the onset/offset stimulus responses of RGCs (S6D and S6I Fig). Finally, RGC RFs did not show significant changes in size, either (S8 Fig).

Although diverse effects of histamine across retinal cell types have been reported in previous ex vivo studies [6,44–49], the histaminergic effects observed in this study were rather uniform across RGC or LGN populations. First, the effects on the peak responses to the contrast-inverting stimuli did not significantly differ between the stimulus onset and offset (Figs 1 and S6). Second, the effects on the temporal filter and nonlinearity were not substantially different among the six functional groups that we identified for RGCs and LGN cells, respectively (S4 and S7 Figs). Our datasets are, however, not large enough to identify all the ~40 physiological cell-types reported thus far [50,51]; and we could not target a specific cell-type or tailor visual stimuli for each cell during recordings due to technical limitations. Lastly, we identified direction-selective (DS) and orientation-selective (OS) cells using the responses to moving grating stimuli in eight different directions (see Materials and methods for details), but we found no differences in the DS or OS index values—defined by the normalized population vector length—before and after each treatment (S9 Fig).

Taken all together, our data indicate that (1) the mouse early visual processing is subject to histaminergic modulation controlled by TMN, starting from the retina; (2) histamine generally suppresses the firing and slows the response kinetics; and (3) response latency and frequency tuning are likely modulated by different mechanisms.

## Pupil dynamics and locomotion are irrelevant to histaminergic modulation of the early visual processing in mice

Histamine is a potent neuromodulator implicated in various brain functions and behaviors, such as sleep–wake cycles and arousal [1,8]. Thus, it is possible that the observed histaminergic effects on the visual response properties in the mouse early visual system are a result of correlated behavioral modulation. For example, a larger pupil size leads to stronger responses because more photons reach the retina [29]. To address this issue, we examined the relationship between the histaminergic effects on the RGC/LGN visual responses and simultaneously monitored behavioral measures (Figs 4 and S10).

We first assessed the effects of the pupil size on the RGC/LGN visual responses. In particular, we performed the reverse correlation analysis using subsampled data with a constricted or dilated pupil for each recording (below 33 or above 66 percentiles of the pupil size, respectively; e.g., Fig 4A–4C). Under the control condition, we found that the larger the pupil size, the faster the response kinetics for both RGCs (Fig 4D and 4E) and LGN cells (S10A and S10B Fig). We also found that the LGN responses were generally stronger with a dilated pupil than with a constricted pupil (S10C Fig). In contrast, variations in the mean evoked firing rate of RGCs between constricted and dilated pupil periods were not uniform

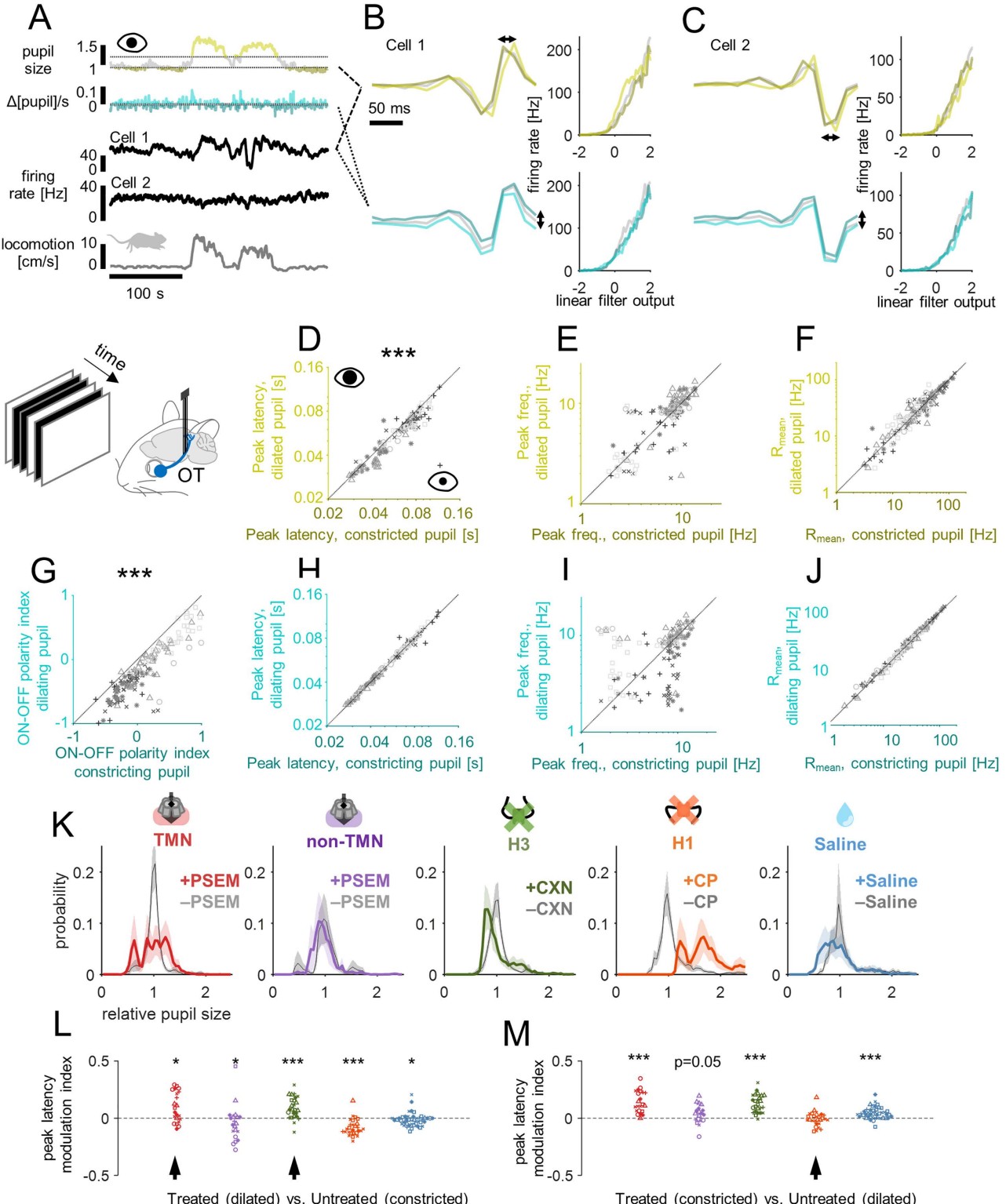

**Fig 4. Histaminergic modulation of pupil dynamics was not causally linked with that of RGC/LGN visual responses. (A)** Representative time series of the pupil size (light yellow, top 33 percentile; dark yellow, bottom 33 percentile) and its time derivative (light blue, positive; dark blue, negative) during randomly flickering full-field stimulus presentation, along with those of the firing rate dynamics of two example RGCs and locomotion (from

top to bottom). **(B)** Estimated linear filter and nonlinearity for a representative RGC (B, cell 1) using constricted or dilated pupil periods (top) or using constricting or dilating pupil periods (bottom). Note a temporal shift of the linear filter in the former case, while a shift in the filter strength in the latter. **(C)** Corresponding data for another RGC (cell 2). **(D–F)** Comparison of the RGC population response properties between constricted and dilated pupil periods ($n = 152$ RGCs from 16 animals): D, peak latency, $65 \pm 19$ ms vs. $60 \pm 17$ ms, median $\pm$ median absolute deviation, $p < 0.001$, Wilcoxon signed-rank test; E, peak frequency, $8.2 \pm 2.6$ Hz vs. $8.2 \pm 2.7$ Hz, $p = 0.6$; F, mean firing rate, $26 \pm 19$ Hz vs. $28 \pm 18$ Hz, $p = 0.6$. **(G–J)** Comparison of the RGC population response properties between constricting and dilating pupil periods: G, ON-OFF polarity index of the temporal filters, $-0.02 \pm 0.33$ vs. $-0.25 \pm 0.33$, $p < 0.001$; H, peak latency, $62 \pm 18$ ms vs. $63 \pm 18$ ms, $p = 0.9$; I, peak frequency, $8.3 \pm 2.8$ Hz vs. $8.3 \pm 3.0$ Hz, $p = 0.9$; J, mean firing rate, $26 \pm 18$ Hz vs. $28 \pm 19$ Hz, $p = 0.7$. **(K)** Probability distributions of relative pupil size before and after treatment: from left to right, PSAM/PSEM for TMN HDC+ cells (red) or non-TMN cells (purple), ciproxifan (green), chlorphenamine (orange), and saline (blue). **(L)** Comparison of peak latencies before treatment with constricted pupil vs. after treatment with dilated pupil. Histamine nevertheless significantly slowed the RGC population responses (arrow heads): * $p < 0.05$; ** $p < 0.01$; *** $p < 0.001$, Wilcoxon signed rank test. **(M)** Comparison of peak latencies before treatment with dilated pupil vs. after treatment with constricted pupil. Effects of H1 blocker could not be distinguished from those of pupil dilation (arrow head, $p = 0.5$). Data and code underlying this figure are available at https://doi.org/10.5281/zenodo.17016431.

across the population, but dependent on their response polarity ($R = 0.20$, $p = 0.015$, Pearson's correlation with the temporal filter ON–OFF polarity index, defined by the difference between the peak and valley, normalized by the sum of the two; $n = 152$ RGCs from 16 animals).

How does this pupil size effect interrelate with the histaminergic effect? As reported before [52], chlorphenamine (H1 antagonist) led to a substantial increase in the baseline pupil size due to its anticholinergic action (Fig 4K; $67 \pm 34\%$, mean $\pm$ standard deviation (SD) of the median pupil size, $n = 6$ animals). We compared the RGC kinetics after blocking H1 with a constricted pupil (hence relatively slower) to those before the treatment with a dilated pupil (hence relatively faster), and found no significant difference (Fig 4M). Thus, we cannot exclude a possible contribution of the baseline pupil dilation to the response facilitation effect of chlorphenamine. In contrast, chemogenetic activation of TMN HDC+ cells resulted in a slight increase in the baseline pupil size ($2 \pm 23\%$, $n = 6$ animals), while ciproxifan (H3 antagonist) treatment led to a slight decrease (Fig 4K; $-7 \pm 18\%$, $n = 8$ animals), though they both resulted in slower RGC/LGN response dynamics (Figs 3 and S6). Such discrepancies are likely due to side effects of the manipulations beyond histamine: e.g., ciproxifan has modest affinity to adrenergic receptors [53] and inhibits monoamine oxidase A and B [54], while some HDC+ cells co-release histamine and GABA [21]. Here, we found that the response kinetics of RGC/LGN after these manipulations with a dilated pupil (hence relatively faster) were nevertheless slower than those under the control condition with a constricted pupil (hence relatively slower; Fig 4L). Therefore, the resulting slow response kinetics after up-regulating histamine cannot be attributable to the pupil size effects.

We next analyzed the effect of pupil dynamics. In general, changes in pupil size occur much more slowly than the response dynamics of the early visual system: e.g., pupillary light reflex has a latency of hundreds of milliseconds and a time-constant of about a second [55–57]. Nevertheless, when the pupil is dilating, presented stimuli on the screen should appear brighter on the retina than intended, allowing RGCs to respond to physically darker stimuli and thereby biasing the response polarity balance. Conversely, one would expect the opposite effect when the pupil is constricting. We indeed identified the corresponding changes in the ON/OFF balance of both RGC and LGN responses (Figs 4G and S10D), using the temporal filter ON-OFF polarity index. An increase of the mean evoked firing rate was also observed in LGN, but not in RGCs, during pupil dilation (Figs 4J and S10G). However, no change in the response kinetics were observed in both RGCs and LGN between dilating and constricting pupil periods (Figs 4H, 4I, S10E, and S10F). The frequencies of saccades or eye blinks did not change, either, after the pharmacological or chemogenetic treatments (S5A and S5B Fig). We thus conclude that the pupil dynamics should have little to do with the observed histaminergic effects on the RGC/LGN visual responses.

Locomotion has been reported to profoundly affect information processing throughout the mouse visual system [26,43,58–60]. When we compared the temporal filter characteristics during running and stationary periods (at a threshold of 2 cm/s), we indeed found that locomotion facilitated visual responses of RGCs and LGN cells (S3D and S3E Fig).

Importantly, we excluded these running periods from the analysis of the histaminergic effects described above. Furthermore, the mice remained stationary most of the time under the head-fixed condition (S3A Fig), and neither the chemogenetic nor the pharmacological treatments resulted in a significant change in either the fraction of time spent running (S3B Fig) or the median running speed (S3C Fig). Thus, locomotion and its effects on RGC/LGN responses are irrelevant to the observed histaminergic modulation, and likely involve different mechanisms.

Taken together, neither pupil dynamics nor locomotion behavior was causally linked with the histaminergic modulation of the mouse early visual system. We conclude that the central histaminergic system modulates the mouse vision, effectively leading to slower and weaker responses.

### Response gain modulation is more strongly correlated with peak latency modulation than with peak frequency modulation

What are the mechanisms underlying the histaminergic modulation of the early visual processing? To address this question, we next took a computational approach to examine the relationship between different aspects of the visual response properties that we identified to be affected by histamine (Fig 5). In particular, we simulated the activity of an integrate-and-fire neuron with Poisson spiking and refractoriness in response to white-noise input stimuli (Fig 5A), and processed the output spike trains in the framework of the linear-nonlinear cascade model to characterize the simulated response properties in the same way as the experimental data (Fig 2B). Here, we varied two model parameters, gain and resting baseline, to examine the effects of their changes on the following four aspects of the simulated responses: spike counts, the peak latency and frequency of the estimated linear temporal filter, and the slope of the estimated static nonlinearity (expressed as half-wave rectification). The rest model parameters were fixed in all simulations, including the underlying linear filter given by a trigonometric function with logarithmic scale in time, and the refractory period filter given by an exponential function. The spike threshold was drawn from a random uniform distribution ranging from 0 to 1 for each time point to approximate the Poisson spiking properties.

When the model neuron had a higher gain or higher resting baseline, subthreshold signal reached the spike threshold faster and more frequently; hence, total spike counts became higher (Fig 5B) and the peak latency of the estimated linear temporal filter appeared shorter (Fig 5C). In contrast, the apparent peak frequency of the simulated responses was dependent more strongly on the resting baseline than on the gain (Fig 5D). These changes in the estimated linear temporal filter properties were consistent with the experimental observation, where the magnitude of the response gain modulation by histaminergic perturbation was significantly correlated with that of the peak latency modulation (Fig 5F and 5H), but much less with that of the peak frequency modulation (Fig 5G and 5I). This agreement between the observed and simulated data indicates that, despite the simplicity, our model captures aspects of neuronal response modulation to a first approximation.

The model analysis supports that the histaminergic effects on RGCs should involve at least gain and baseline modulations. First, the H1 receptors likely mediate the gain modulation in RGCs, given that chlorphenamine (H1 antagonist) led to changes in the peak latency but not the peak frequency, along with changes in the estimated static nonlinearity largely by scaling (Fig 3). Second, the peak frequency was modulated as well as the peak latency of RGCs upon chemogenetic activation of HDC+ neurons in TMN, accompanied by a shift of the estimated static nonlinearity. This implies changes in the baseline resting potential of the cells. Blocking the H3 receptors, however, led to changes in the RGC response kinetics without much affecting the mean firing rates. Furthermore, the net effects of histamine perturbations on LGN responses were more diverse (Figs 1 and 2) and not fully explainable by the simple model (Fig 5). Thus, we expect that mechanisms besides gain and baseline modulations are also involved, such as neural circuit effects through retinal amacrine cells for example [48].

## Discussion

Here we provide in vivo electrophysiological evidence for histaminergic modulation of the mouse early visual system: the more the histamine released, the slower and weaker the LGN and RGC visual responses in awake head-fixed mice (Figs

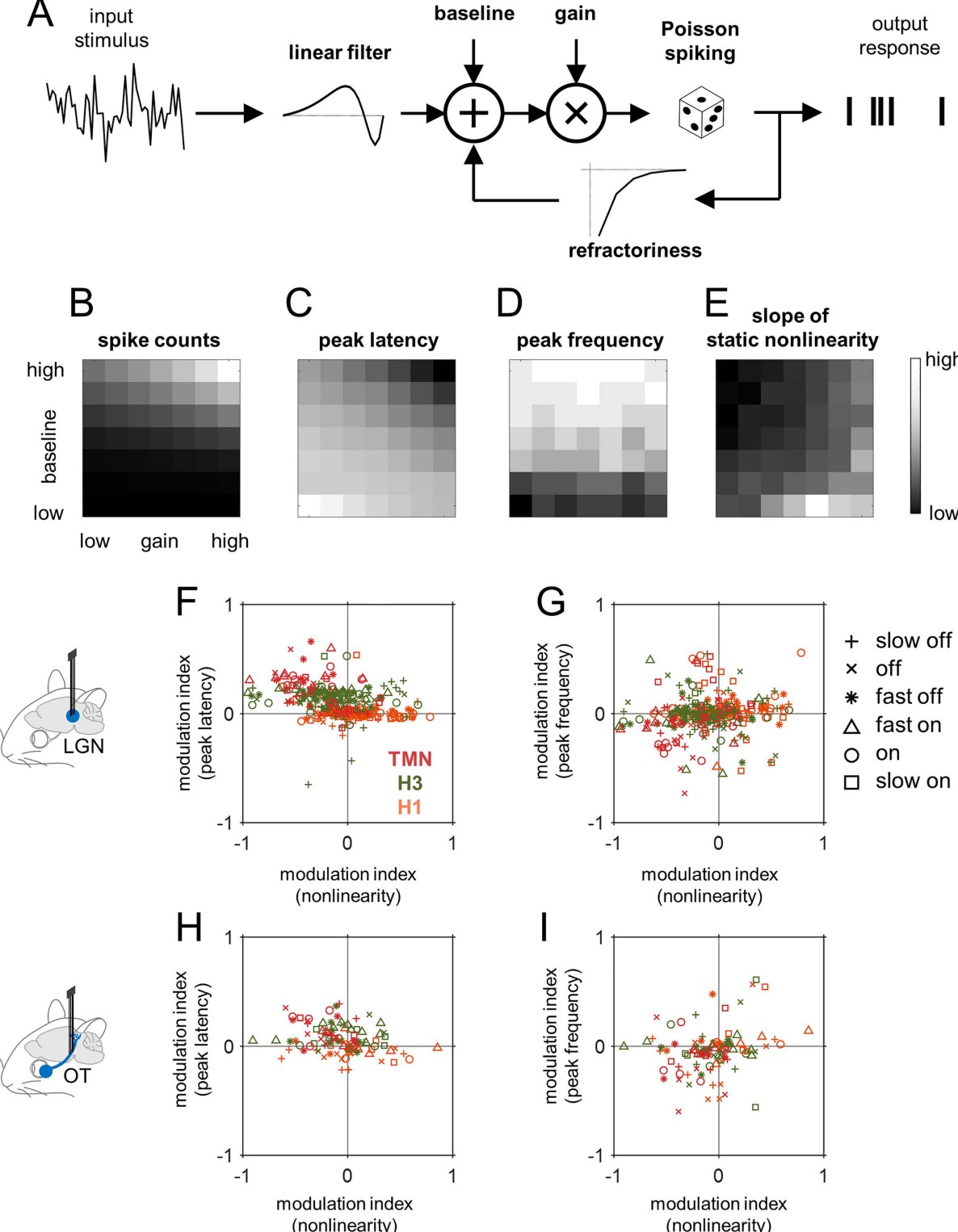

**Fig 5. Gain modulation accompanies the peak latency modulation but not the peak frequency modulation via histaminergic effects. (A–E)**
Using a computational model outlined in A, here we examined the consequences of varying the baseline and gain levels on the response features, such as spike counts (B), peak latency (C), and frequency (D) of the estimated linear filter from the simulated responses, and slope of static nonlinear

gain function of the model neuron (E). See Materials and methods for details. **(F,G)** Relationship between the magnitudes of the gain modulation and response kinetics modulation by perturbing the histaminergic system for LGN neurons (chemogenetics, red; ciproxifan, green; chlorphenamine, orange; $n = 300$ from 10 animals). Cell types are indicated by different markers (S4 and S7 Figs). A stronger correlation was found for the peak latencies (F, $R = -0.35$, $p < 0.001$) than for the peak frequencies (G, Pearson's $R = 0.12$, $p = 0.18$). Here, we combined all data across cell types and experimental conditions to assess the relationship between the two dependent variables as a result of histamine level manipulations. **(H,I)** Corresponding figure panels for RGC responses ($n = 101$ from 11 animals). Significant correlation to the gain modulation was found for the peak latency modulation ($R = -0.26$, $p = 0.03$), but not for the peak frequency modulation ($R = 0.22$, $p = 0.11$). Data and code underlying this figure are available at https://doi.org/10.5281/zenodo.17016431.

1–3). Anterograde tracing of TMN HDC+ neurons confirmed the projection of their axons into the optic chiasm and optic nerve, but not in the retinal tissue (S2C–S2E Fig). Nonetheless, here we used the same mouse lines as previous studies that demonstrated the presence of histaminergic HDC+ axons in the retina [5,6]. Moreover, we observed modulation at the level of RGC axons in the OT in vivo (Figs 3 and S6). It is intriguing that punctate signals of presynaptic markers expressed in TMN HDC+ cells were identified in the optic chiasm (S2C Fig), raising a potential role for axonal modulation. However, its contribution should be minimal, if any, because RGC axon potentials should reach the recording site in the OT within 1 ms after exiting the eye cup (optic nerve length in mice, <10 mm; conduction velocity, around 10 m/s) [61]. While axonal conduction velocity depends on various factors, including myelination and channel density, its modulation should then occur only on a much shorter time scale than the observed changes in RGC/LGN response kinetics (up to tens of milliseconds). Taken together, we suggest that the histaminergic modulation likely arises in the retina.

Previous ex vivo studies have reported diverse histaminergic effects in all five major cell types in the retina across species [6,44–49]. Primate RGCs, for instance, either increase or decrease their baseline activity, while their visual responses typically become weaker and slower after histamine application ex vivo [44,46]. In contrast, rodent RGCs mostly show an increase in the baseline activity, with various changes in their visual response properties in a histamine dose-dependent manner [6,44]. Some of those changes are interrelated: e.g., a broadening of direction-selectivity (DS) tuning can be explained as a natural consequence of the increased baseline activity because a decrease of the DS index (DSI) should follow by definition. Indeed, a reduction of DSI was reported at RGC axon terminals, following an increased baseline activity via arousal or locomotion [26,27]. However, the mechanisms underlying the histaminergic modulation of the retina remain mostly unclear.

We took a computational modeling approach to better understand the net effect of histamine from functional viewpoints (Fig 5), rather than pursuing the underlying biophysical mechanisms. For example, we suggest that the H1 receptors mediate gain modulation in RGCs: chlorphenamine (H1 antagonist) affected the response latency but not the frequency tuning (Fig 3), a distinctive feature of gain modulation identified by our computational modeling analysis (Fig 5). This action of histamine does not have to be directly on RGCs expressing the H1 receptors, but can be mediated indirectly by circuit mechanisms. Indeed, histamine affects both inward and outward currents of amacrine cells in the mouse retina [48]. The effects of histamine can then be more profound than gain modulation, much as we observed diverse outcomes in RGCs and LGN when different pharmacological and chemogenetic perturbation tools were employed (see also [6]).

Correlation between histamine levels and various behavioral measures has been widely reported [62,63]. For instance, the activity of central histaminergic neurons generally follows the circadian rhythm, showing high levels of activity during the active period (i.e., night time for mice as nocturnal animals) and low levels during the sleep period [7,10]. Here we performed all our experiments during daytime (ZT4-ZT9). Moreover, the mice remained calm under head fixation, as evidenced by their low spontaneous running behavior (S3 Fig), following a week of habituation sessions before recordings. The retinal histamine level of the subject mice was then expected to be relatively low during the recordings, given that the mammalian retina contains a comparable level of histamine with other brain areas (around 20–140 ng/g wet tissue, i.e., about 0.1–1 μM) [64]. This will explain why we found stronger effects by chemogenetically activating HDC+ neurons in TMN or pharmacologically blocking H3 receptors (mimicking an increase of histamine) than by administering

H1 antagonists (mimicking a decrease of histamine). However, regulation of the histamine release and metabolism is suggested to be both tissue- and species-specific [64]: e.g., light stimulation of dark-adapted rabbits at night leads to a substantial reduction of histamine in the retina and the optic nerve (by about 40%–50%), but not in other parts of the central nervous system. It is a future challenge to determine the exact level of retinal histamine and its fluctuation under more precisely controlled behavioral contexts.

Our observations on the histaminergic effects have certain discrepancies from those reported in previous studies. For example, it has been reported that histamine increased the baseline activity of RGCs in the mouse retinal explants [6]. This effect was dose-dependent and saturated at 5 μM histamine with an increased baseline firing by ~4 Hz on average. We instead found a reduced firing upon H3 antagonist application or chemogenetic activation of TMN HDC+ cells, while an enhanced firing after blocking H1 receptors (Figs 3 and S6). This might be because the baseline activity of RGCs in awake mice is already high, by ~20 Hz compared to the ex vivo condition [29]. Alternatively, the enhancement of the baseline activity in retinal explants could be an artifact of histamine overdose. An isolated retinal tissue should have no baseline histamine because the retina itself does not produce histamine [3,5]. However, bath-application of histamine at the physiological range (0.1–1 μM) [64] or that of H3 antagonists had no effect on the baseline firing ex vivo [6]. Moreover, chemogenetic activation of HDC+ axon terminals in retinal explants led to a net reduction of firing, with ~10% of RGCs showing a significant decrease of the baseline firing while only ~1% showing an increase [65]. The retinal environment clearly differs between ex vivo and in vivo conditions. One should then be cautious as findings from ex vivo studies may not be directly applicable to the awake, more natural state.

What is the biological relevance of histaminergic modulation of the early visual system, starting at the retina? We speculate that species-specific ethological factors play a role. For example, some species may sleep with their eyes open [66–69], including mice [70–73]. Furthermore, as nocturnal animals, mice are less active during daytime. Facilitating visual responses at low histamine level may then be ethologically beneficial for mice to respond faster to visual threats during daytime, especially when mice are less active. In contrast, when animals are more active, other mechanisms, such as arousal, may compensate for the histaminergic effects [26,27]. Such arguments also hold for those species without eyelids, including some amphibians, reptiles, and fish. To our knowledge, we are the first to report histaminergic effects on LGN in awake mice. Recordings from LGN in cats and guinea pigs (under the anesthetized condition or brain slices [74,75]) suggest that histamine leads to a slow depolarization of the relay cells, enhancing the baseline activity but introducing a lag in their visual responses. However, substantial differences exist in the early visual processing between awake and anesthetized conditions [29,76]. Further studies are needed to address how histamine affects visual processing and perception in awake animals in tissue- and species-specific manners.

Growing evidence supports significant correlations between awake retinal responses and various behavioral measures, such as pupil dynamics and locomotion [26–29]. Such correlations were also found in our data sets; however, we demonstrated that an animal's pupil dynamics and locomotion behavior have little to do with the observed histaminergic effects on RGC/LGN responses (Figs 4, S3, and S10). This indicates that mechanisms other than histamine are involved in the retinal modulation by arousal and locomotion. The retina exploits not just histamine but also many other neuromodulators, synthesized inside [77] or outside the retina [36,37], such as the centrifugal serotonergic system [78–80]. It is a future challenge to identify what behavior is controlled by histamine and vice versa, and characterize the anatomical and physiological features of those HDC+ cells projecting to the retina to clarify behavioral relevance of the histaminergic modulation of the early visual system.

## Materials and methods

No statistical method was used to predetermine the sample size. The significance level was 0.05 (with multiple comparison correction where appropriate) in all analyses unless noted otherwise. All experiments were performed under the license 233/2017-PR and 220/2024-PR from the Italian Ministry of Health, following protocols approved by the Institutional

Animal Care and Use Committee at European Molecular Biology Laboratory. The data analyses were done in Python and Matlab.

## Animals

Animals were housed on a 12 h light-dark cycle, with ad libitum access to water and food. All experiments were performed on female mice. A total of 22 wild-type (C57BL6/J; RRID:IMSR_JAX:000664) and 12 hemizygous HDC-Cre (Hdctm1.1(i-cre)Wwis/J; RRID:IMSR_JAX:021198) mice were used for pharmacological and chemogenetic manipulations, respectively. Additional 9 hemizygous HDC-Cre mice were used for anterograde tracing of HDC+ cells in TMN.

## Surgical procedures

All surgical procedures were performed in animals from 4 to 19 weeks old (9.2 weeks median age). Before the surgery, animals were injected with Carprofen (5 mg/kg) and then anaesthetized with isoflurane (4% for induction, 1% for maintenance in $O_2$). Throughout the surgery, the temperature of the animals was kept stable at 37 °C using a heating pad (Supertech Physiological). Ointment (VitA-Pos, Ursapharm) was applied on both eyes to prevent them from drying during the surgery, and the animals were placed in a stereotaxic frame (Stoelting 51625). A portion of the scalp was removed to expose the skull, and the periosteum was scraped away with a round scalpel to increase adherence of the dental cement. For those mice belonging to the chemogenetic manipulation group, the skull was aligned to level bregma and lambda on the same horizontal plane, and a 100 μm craniotomy was performed using a motorized drill attached to the stereotaxic arm. Viral solution (250 nL; AAV9::FLEX-PSAM Y115F,L141F:5HT3 HC; Vector Biolabs, VB773) was injected with an automatic injector pump at 1 nL/sec, at [−2.6, ±0.8, −5.3] or [−1.7, ±0.5, −5.3] in [Anterior–Posterior (AP), Medial–Lateral (ML), Dorsal–Ventral (DV)] coordinates for targeting TMN and non-TMN areas of the hypothalamus, respectively. All animals were then implanted with a custom-made titanium head plate, cemented on the skull using a mixture of cyanoacrylate (Loctite 401, Henkel) and dental cement (Paladur, Kulzer). The head plate featured a hole (8 mm diameter), allowing for clear observation of both bregma and lambda. The skull surface was then glazed with a thick layer of cyanoacrylate to support the skull with mechanical, atmospheric, and biological protection, while still allowing for visual identification of reference points. After the surgery animals were allowed to recover on a heating pad and then housed in individual cages. During the following seven days, the mice were administered with analgesia (Carprofen; 50 mg/mL) diluted in the drinking water.

## Anterograde tracing

For anterograde tracing of HDC+ cells, we injected recombinant AAVs carrying Cre-dependent fluorescent markers (100 nL; rAAV9::CAG-FLEX-Synaptophysin-GFP or AAV2/1::CAG-FLEX-axon-GFP) into the TMN of HDC-Cre mice ($n = 9$) as described above. At least 10 days after the injection, the animals were anesthetized (2.5% Avertin, 16 μL/g, intraperitoneal injection) and perfused with paraformaldehyde (PFA; 4% in phosphate buffer solution). The brain tissues including the optic chiasms and optic nerves were dissected and post-fixed overnight in 4% PFA at 4 °C. Coronal sections of the brain tissue (thickness, 80 μm) were then examined under a laser scanning confocal microscope (Leica TCS SP5; S2 Fig).

## In vivo electrophysiology

After recovery from the surgery, animals were habituated to head fixation by placing them in the experimental apparatus over the course of one week, twice a day for up to two hours. On the day of the recording, we first placed the subject animal in the recording set up with its head fixed, and determined the electrode penetration path to the target area (OT, [−1.34, +1.87, +4.74], [−1.70, +1.87, +4.74], or [−1.82, +2.35, +4.07] in [AP, ML, DV] coordinates; LGN, [−2.3, +2.3, +2.8]) using the robotic stereotaxic system (StereoDrive, NeuroStar). The animal was then briefly anesthetized (with isoflurane

for about 5 min) and a hole was drilled around the electrode entry point on the skull. After the removal of the anesthesia, an acute silicone probe (Buzsaki32L, Neuronexus; or P2, Cambridge Neurotech) coated with a fluorescent dye (DiI stain, Invitrogen, D282) was lowered at 5 μm/s using the robotic arm until visual responses were found in the target area. All recordings were done during the daytime (between 11 am/ZT4 and 4 pm/ZT9; 1 pm/ZT6±1 h, mean±SD; 96% of the paired recordings were conducted within 2 h of each other).

After presenting a battery of visual stimuli, we briefly anesthetized the animal with isoflurane and intraperitoneally injected 0.9% saline solution alone or that containing chlorphenamine (H1 antagonist, 5 mg/kg) [39,40], ciproxifan (H3 antagonist, 12 mg/kg) [41,42], or PSEM (5 mg/kg) [38]. About 20 min after the injection and removal of the anesthesia, we presented the same battery of visual stimuli to the animal for the second recording session. Here, we did not target the H2 or H4 receptors because blocking the H2 receptors has little effect on retinal physiology ex vivo [6], and the functional role of the H4 receptors in the central nervous system remains unclear [9]. In our electrophysiology experiments, we could not keep the same cells long enough to test if the effects of these manipulations were reversible or not.

After all the recording sessions, the electrode position was verified histologically (e.g., Figs 1B, LGN and 3B, OT). After retracting the silicon probe, the mice were anesthetized (2.5% Avertin, 16 μL/g, intraperitoneal injection) and perfused with 4% PFA. The brain tissue was harvested and post-fixed overnight in 4% PFA at 4 °C. Coronal sections of the brain tissue (thickness, 100–150 μm) were then examined under a fluorescence microscope (Leica, LMD7000 with N2.1 filter cube) to visualize the trace left by the DiI stain on the probe. To examine the PSAM expression, staining of the slices was performed (S2A, and S2B Fig): primary staining, 1-h incubation with an alpha-bungarotoxin/biotin-XX conjugate (ThemoFisher, B1196) diluted with PBS with 0.1% Tween (1/500) at room temperature; secondary staining, streptavidin and alexa-fluorophore, 1/500 in PBS(T) 0.1% for 2 hours at room temperature.

## Visual stimulation

Visual stimuli were presented as described before [29]. Briefly, visual stimuli were projected onto a spherical screen (20 cm in radius) painted with ultra-violet (UV) reflective white paint, placed 20 cm from the mouse's left eye. A gamma-corrected digital light processing device (DLP, Texas Instruments, DLPDLCR3010EVM-LC) was used as a light source after the green and the red light-emitting diodes (LEDs) were replaced with UV (365nm, LZ1-00UV00, LED Engine) and infrared (IR; 950nm, SFH 4725S, Osram) LEDs, respectively. The IR light was used as a synchronization signal and recorded with a photodiode (PDA100A2, Thorlabs). Stimulation was conducted at 60 frames per second (FPS), and covered 73° in azimuth and 44° in altitude from the mouse eye position. The maximum light intensity at the eye position was 31 mW/m$^2$ (15.4 mW/m$^2$ for UV LED and 15.9 mW/m$^2$ for blue LED; measured with S121C sensor, Thorlabs), leading to mesopic to photopic conditions. We presented the following stimuli using QDSpy: a randomly flickering full-field stimulus (5 min), consisting of a random sequence of black and white frames at 60 FPS; a black-and-white binary dense noise stimulus (15 min), consisting of a 32-by-18 pixels checkerboard, where each pixel followed an independent random sequence at 60 FPS but the overall luminance of the frame was kept constant at the mean value; a moving grating stimulus in eight different directions (spatial frequencies of square waves, 3° or 20°; moving speed, 7.5°/s or 15°/s); full-field contrast-inverting stimuli in an "OFF-ON-OFF" sequence at maximum contrast (2 s each), followed by a sinusoid (1.5 Hz) with a linearly increasing amplitude from 0% to 100% contrast over 10 s (10 trials) or a sequence of sinusoids with different temporal frequencies (1.875, 3.75 and 7.5 Hz, each for 2 s; 15 and 30 Hz for 1 s) at maximum contrast (10 trials), with an inter-trial interval of 1 s (gray screen).

## Electrophysiology data analysis

We adapted previously established methods of spiking sorting and OT data analysis [29]. In brief, we first concatenated the raw data from the same animal before and after treatment into a single binary file, and used Kilosort 2.0 to sort spikes with a set of default parameters, except for the spike detection threshold to be 6 during optimization. Single units

were then identified by clustering in principal component space using Phy for visualization and manual data curation. Only those units that maintained the average spike waveforms and autocorrelograms with a minimal refractory period of 1 ms were kept for subsequent analyses (S1 Fig). Specifically, we selected RGCs/LGN cells as those with robust visual responses: i.e., $\text{SNR} = \text{var}\left[\langle r(t) \rangle\right]_t / \langle \text{var}\left[r(t)\right]_t \rangle > 0.15$, where $r(t)$ is the response during the "ON-OFF" part of the OFF-ON-OFF stimulus sequence, $\langle \cdot \rangle$ indicates the mean over trials, and $\text{var}[\cdot]_t$ the variance over time $t$ (bin size, 1/60 ms), respectively; and the estimated temporal filter should have $p < 10^{-10}$ in at least one time bin within 200 ms from the spike onset (see below for details). Single-units with little or no visual responses were excluded from the analysis as nontarget cells, such as the axons from the parabigeminal nucleus in the OT [81,82]. In total, we obtained 337 RGCs from 20 animals and 557 cells in LGN from 14 mice. These RGCs typically had a triphasic spike waveform as expected for axonal signals [83].

To characterize the peak latency and firing rate of the responses to full-field contrast-inverting stimuli (Figs 1 and S6), we first identified the cell's response preference to stimulus polarity using an ON–OFF index defined as $(r_{\text{ON}} - r_{\text{OFF}})/(r_{\text{ON}} + r_{\text{OFF}})$, where $r_{\text{ON}}$ and $r_{\text{OFF}}$ are the mean firing rate during the ON and the second OFF periods of the OFF–ON–OFF stimulus sequence. We then computed the peri-stimulus time histogram (bin size, 1 ms), smoothed it with a Gaussian filter (kernel width, 6 SD), and identified the first peak upon stimulus onset or offset for those cells with positive or negative ON–OFF index values, respectively. We then performed pairwise comparisons on the peak latencies and firing rates before and after each treatment (e.g., Fig 1F–1J). To compare the changes in the detected peak latency and firing rates across different conditions (e.g., Fig 1K and 1L), we performed Kruskal–Wallis test on a modulation index defined as $\frac{(a_{\text{after}} - a_{\text{before}})}{(a_{\text{after}} + a_{\text{before}})}$, where $a$ is either the detected peak latency or peak firing rate before and after treatment.

For systematically characterizing the visual response properties, we used stimulus ensemble statistical techniques (reverse correlation methods; 500 ms window; $\Delta t = 1/60$ s bin size) to calculate the linear filter and static nonlinear gain function of the recorded cells in response to white-noise stimuli as described before (Fig 2B) [29]. Briefly, we first obtained the linear filter of each cell by calculating a STA of the stimulus with ±1 being "white" and "black," respectively, under a given behavioral state (e.g., when the animal stayed stationary below 2 cm/s). As a quality measure, $p$-value was computed for each time bin against a null hypothesis that the STA follows a normal distribution with a mean of zero and a variance of $1/C$, where $C$ is the total number of spikes. As a measure of the cell's response kinetics, we then estimated the peak latency by fitting a difference-of-Gaussian curve to the linear filter; and the spectral peak frequency by the Fourier analysis on the linear filter. The ON–OFF polarity index of a temporal filter was defined as the difference of its peak and valley, divided by the sum of the two (Figs 4G and S10D). Spatial response properties were examined for STAs from "checkerboard" stimulation, using the time window where the eye position remained within 1 inter-quartile range in both horizontal and vertical coordinates for at least 2 s (S5C and S8A Figs) to minimize the effects of eye movements while retaining enough data for RF estimation. In particular, we fitted a two-dimensional Gaussian envelope to the spatial filter at the peak latency (e.g., S5D Fig), and the RF size was estimated as twice the mean SD of the long and short axes. Modulation index was used to characterize the change in a response feature before and after treatment, such as the peak latency and frequency, the mean firing rate, and the RF size (Figs 2K–2N and 3K–3N). Here we focused only on those data when the mouse stayed stationary (<2 cm/s) to minimize the side effects of locomotion behavior (S3 Fig) [26,43].

The linear temporal filters from full-field white-noise stimuli were used for broadly classifying the response types of RGCs and LGN cells, respectively (S4A, S4B, S7A, and S7B Figs). Specifically, we first used t-Distributed Stochastic Neighbor Embedding to map the STAs onto two-dimensional space, and then used K-means++ algorithm for heuristically categorizing the responses into the following six types: fast ON, ON, slow ON, fast OFF, OFF, and slow OFF. Our data sets were not large enough to physiologically classify and identify all the cell types reported thus far [50,51]. Direction-selectivity (DS) and orientation-selectivity (OS) indices were calculated by projecting the responses to the moving grating stimuli in eight different directions onto a complex exponential: $\text{DS/OSindex} = \left\| \frac{\sum_k e^{-i\alpha\omega_k}}{\sum_k r_k} \right\|$, where $\omega_k$ and $r_k$ are the angle of the $k$-th direction and the cell's corresponding responses, respectively; and $\alpha = 1$ and 2 for the DS and OS indices, respectively (S9 Fig).

Static nonlinear gain function of each cell (bin size, 0.1) was computed as: $P$(response | stimulus) = $N$(stimulus | response)/$N$(stimulus)/$\Delta t$, where $N$(stimulus | response) and $N$(stimulus) are the distributions of spike-triggered stimulus ensembles projected onto the L2-normalized STA and the entire stimulus ensembles, respectively. The ratio of $P$(response | stimulus) before and after the treatment was used as a measure of response gain modulation for each cell.

## Behavioral data analysis

For all experiments, the animal's left eye—i.e., the side presented with visual stimuli—was recorded using an IR camera (The Imaging Source, DMK23UV024) to monitor its motion and pupil dynamics at 30–60 Hz. Pupil detection was done using a Mask-Region based Convolutional Neural Network, trained as described previously [29]. A two-dimensional ellipse was fit to the pupil, and the large axis diameter was used as a measure of the pupil size, normalized by the median value before treatment for each animal (Fig 4A). To examine the pupil size effects on the response dynamics, a threshold at 33 and 66 percentiles was used to define constricted, neutral, and dilated pupil period, respectively. To examine the effects of the pupil dynamics, we took the time derivative of the pupil size dynamics, and set a threshold at ±0.01 point per second to define constricting, stable, and dilating pupil periods, respectively.

We also monitored the running speed of an animal based on the turning speed of the custom-made rotary treadmill throughout the recordings. We set a threshold at 2 cm/s to detect running behavior (S3A Fig). For quantification, we measured the fraction of the running period (S3B Fig), and the median running speed when mice moved at >2 cm/s before and after each treatment (S3C Fig).

## Model analysis

To explore how different aspects of neuronal intrinsic properties and output responses are interrelated between each other, we simulated a model neuron's responses to white-noise stimuli using different model parameter values (Fig 5A), and characterized these simulated responses using a linear-nonlinear cascade model as we did for the experimental data.

Specifically, we employed an integrate-and-fire neuron model with Poisson spiking and refractoriness. A model neuron's discrete linear filter (of length $n_T = 25$) was given by a trigonometric function with logarithmic scale in time: i.e., $L(i) = \sin\left[\frac{2\pi\left(10^{t_i}-1\right)}{(10^{s_t}-1)}\right]$, where $s_t = 1.5$ is a temporal spacing parameter and $t_i = \frac{i \cdot s_t}{(n_T-1)}$ for $i = 0,\ldots,n_T-1$. The spike-generating function $V(i)$ was then modeled by first projecting the white-noise Gaussian stimulus $S(i)$ (of length 1,000,000; normalized to have a maximum value of 1) onto $L(i)$, followed by the addition of baseline $B$ and the multiplication by gain $G$: i.e., $V(i) = G \cdot \left(\sum_j L(j) \cdot S(i-j) + B\right)$. For each simulation, we chose the value of $B$ from −0.45 to +0.45 in steps of 0.15; and that of $G$ from $1.2^g$ with $g = -3,\ldots,+3$. The spiking response was then generated by thresholding $V(i)$ (at a random threshold for every time point derived from a uniform distribution from 0 to 1) and refractoriness (given by an exponential filter: $-\exp\left[\frac{-i}{s_h}\right]$ for $i = 0,\ldots,5$ with $s_h = 1$). The output spike trains were then subject to the reverse-correlation analysis as described above to characterize the apparent linear filter and static nonlinearity of the model neuron, much as we did for the experimentally recorded RGCs and LGN cells. Specifically, we focused on the peak latency and frequency of the estimated linear filters and the slope of the estimated static nonlinear gain function, and examined how these features depended on the baseline $B$ and the intrinsic gain $G$ of the model neuron to provide a minimal phenomenological explanation to our observations (Fig 5B–5E).

## Supporting information

**S1 Fig. Spike waveform and autocorrelation of spike trains of representative cells across different conditions. (A)** Average spike waveform (top) and auto-correlogram (bottom) of two representative LGN cells (left and right) before (black) and after (red) chemogenetic activation of TMN HDC+ cells. **(B–D)** Corresponding data for ciproxifan (B, green), chlorphenamine (C, orange), and saline administration (D, blue), each with two representative LGN cells. **(E–I)**

corresponding data for representative RGCs from optic tract recordings (E, chemogenetic activation of HDC+ cells in TMN; F, chemogenetic activation of HDC+ cells outside TMN; G, ciproxifan; H, chlorphenamine; I, saline). Data and code underlying this figure are available at https://doi.org/10.5281/zenodo.17016431.
(TIF)

**S2 Fig. Pharmacological and chemogenetic approaches to manipulate the central histaminergic system in the mouse brain. (A,B)** Viral delivery of PSAM 5HT3HC channel to HDC+ cells in the TMN (marked with dotted lines) of the posterior hypothalamus (A) or those in the anterior hypothalamus (B; control). **(C–E)** Histological image examples of anterograde tracing of HDC+ cells in TMN. Labeled axons were found in the optic chiasm (C) or the optic nerve (D) after injecting rAAV9::CAG-FLEX-Synaptophysin-GFP or AAV2/1::CAG-FLEX-axon-GFP, respectively, in the TMN of HDC-Cre mice. However, no visible signal was detected in the isolated retinal tissue of all animals examined ($n = 9$; see panel E for example). Data and code underlying this figure are available at https://doi.org/10.5281/zenodo.17016431.
(TIF)

**S3 Fig. Locomotion of an animal facilitates the visual responses of LGN and RGCs. (A)** Probability distribution of an animal's locomotion speed during recordings before (gray) and after treatment: from left to right, chlorphenamine (CP, orange), ciproxifan (CXN, green), PSAM/PSEM for HDC+ cells in non-TMN (purple) or TMN (red), and saline (blue). **(B)** The fraction of running period (>2 cm/s) before and after each treatment. No significant change was observed ($p = 0.07$, Kruskal-Wallis test). **(C)** The median locomotion speed (during which animals moved at >2 cm/s) before and after each treatment. No significant change was observed ($p = 0.3$). **(D)** Comparison of the LGN population response properties between stationary and running periods ($n = 104$ from 5 animals with a running period ranging between 20% and 80%): from left to right, peak latency ($50 \pm 19$ ms versus $48 \pm 19$ ms, median $\pm$ median absolute deviation; $p < 0.001$, Wilcoxon signed-rank test), peak frequency ($8.2 \pm 1.9$ Hz versus $8.2 \pm 3.0$ Hz; $p = 0.4$), mean firing rate ($11 \pm 6$ Hz versus $20 \pm 8$ Hz; $p < 0.001$). **(E)** Corresponding data for RGCs ($n = 44$ from 5 animals with a running period ranging between 20% and 80%): from left to right, peak latency ($58 \pm 19$ ms versus $49 \pm 18$ ms; $p < 0.001$), peak frequency ($8.2 \pm 2.8$ Hz versus $9.2 \pm 2.7$ Hz; $p = 0.6$), mean firing rate ($24 \pm 22$ Hz versus $23 \pm 22$ Hz; $p = 0.5$). Data and code underlying this figure are available at https://doi.org/10.5281/zenodo.17016431.
(TIF)

**S4 Fig. Functional classification of LGN visual response types and batch effect analysis. (A)** t-Distributed Stochastic Neighbor Embedding (t-SNE) embedding of the STAs of LGN cells. Different markers and shadings are used for distinct response categories (shown in B). **(B)** Each panel represents one of the six response categories: slow off, off, fast off, fast on, on, and slow on. In each panel, each row represents a cell's STA (color-coded with red and blue hue, indicating positive and negative filter values, respectively); and the overlaid gray line shows the average STAs in each response type. **(C–F)** Modulation indices on LGN response characteristics across individual animals: from left to right, peak latency (C), peak frequency (D), mean evoked firing rate (E), and nonlinearity (F). The effects of histamine were largely consistent across animals. Data and code underlying this figure are available at https://doi.org/10.5281/zenodo.17016431.
(TIF)

**S5 Fig. Histamine did not affect LGN RF size. (A, B)** Blink (A) and saccade (B) frequencies before and after different treatments: from left to right, chlorphenamine (CP), ciproxifan (CPN), PSAM/PSEM for non-TMN HDC+ cells and TMN HDC+ cells, and saline. None of these changes were statistically significant: $p = 0.10$, Kruskal–Wallis test on blink frequency changes; $p = 0.15$, saccade frequency changes. **(C)** Representative time series of the eye position (X- and Y-coordinates of the pupil center extracted from eye-tracking camera images) during white-noise "checkerboard" stimulation (black/red, centered and stable eye position used for reverse correlation; gray, non-centered or non-stable period excluded from the analysis; probability distribution shown on the right), before (left) and after (right) chemogenetic

activation of TMN HDC+ cells. **(D)** Estimated spatial filter (receptive field; RF) of two example LGN cells before (top) and after (bottom) chemogenetic activation of HDC+ cells in TMN. Note a shift of RF position of all recorded cells due to a shift of the animal's resting eye position after the treatment (see panel C). **(E–H)** Cumulative distribution of the modulation index of LGN cells before and after treatment (E, receptive field size; F, peak latency; G, peak frequency; H, mean evoked firing rate): orange, chlorphenamine, $n = 32$ from 3 animals; green, ciproxifan, $n = 61$ from 4 animals; red, PSAM/PSEM for HDC+ cells in TMN, $n = 20$ cells from 3 animals. Post-hoc test against saline control ($n = 47$ from 3 animals) after Kruskal–Wallis test: * $p < 0.05$; ** $p < 0.01$; *** $p < 0.001$. Data and code underlying this figure are available at https://doi.org/10.5281/zenodo.17016431.
(TIF)

**S6 Fig. Histamine primarily modulated the visual response kinetics of RGCs. (A–E)** Visual responses of a representative RGC to full-field contrast-inverting stimuli (2 s intervals) before and after chemogenetic (A, HDC+ cells in TMN; B, HDC+ cells in non-TMN) or pharmacological treatment (C, ciproxifan; D, chlorphenamine; E, saline); top, spike raster across trials; bottom, zoom-in of the spike raster around stimulus onset or offset (−50 to 250 ms, red shade on top) and peri-stimulus time histogram. **(F–J)** Pairwise comparison of the RGC population responses before and after treatment (top, peak latency; bottom, peak firing rate): * $p < 0.05$; ** $p < 0.01$, Wilcoxon signed-rank test with Bonferroni correction. F, PSAM/PSEM for TMN HDC+ cells: $57 \pm 15$ ms versus $66 \pm 17$ ms peak latency, $p = 0.005$; $122 \pm 54$ Hz versus $112 \pm 66$ Hz peak frequency, $p = 0.08$; median $\pm$ median absolute deviation, $n = 16$ RGCs from 4 animals. G, PSAM/PSEM for non-TMN HDC+ cells: $42 \pm 17$ ms versus $43 \pm 14$ ms peak latency, $p = 0.3$; $46 \pm 30$ Hz versus $59 \pm 40$ Hz peak frequency, $p = 0.3$; $n = 24$ RGCs from 3 animals. H, ciproxifan: $42 \pm 31$ ms versus $53 \pm 31$ ms peak latency, $p = 0.002$; $72 \pm 80$ Hz versus $78 \pm 78$ Hz peak frequency, $p = 0.3$; $n = 32$ RGCs from 4 animals. I, chlorphenamine: $61 \pm 16$ ms versus $47 \pm 18$ ms peak latency, $p = 0.026$; $55 \pm 60$ Hz versus $102 \pm 88$ Hz peak frequency, $p = 0.05$; $n = 22$ RGCs from 3 animals. J, saline: $73 \pm 30$ ms versus $73 \pm 31$ ms peak latency, $p = 0.5$; $92 \pm 50$ Hz versus $92 \pm 56$ Hz peak frequency, $p = 1$; $n = 37$ RGCs from 3 animals. **(K,L)** Cumulative distributions of the modulation index before and after each treatment (in corresponding colors): K, peak latencies; L, peak firing rate; * $p < 0.05$ from the post-hoc test against the saline (control) condition on the average group ranks (Kruskal–Wallis test). Data and code underlying this figure are available at https://doi.org/10.5281/zenodo.17016431.
(TIF)

**S7 Fig. Functional classification of RGC visual response types and batch effect analysis. (A)** t-SNE embedding of the RGC STAs. Different markers and shadings are used for distinct response categories (shown in B). **(B)** Each panel represents one of the six response categories: slow off, off, fast off, fast on, on, and slow on. In each panel, each row represents a cell's STA (color-coded with red and blue hue, indicating positive and negative filter values, respectively); and the overlaid gray line shows the average STAs in each response type. **(C–F)** Modulation indices on RGC response characteristics across animals: from left to right, peak latency (C), peak frequency (D), mean firing rate (E), and nonlinearity (F). The effects of histamine were generally consistent across animals, and no substantial batch effect was observed. Data and code underlying this figure are available at https://doi.org/10.5281/zenodo.17016431.
(TIF)

**S8 Fig. Histamine did not affect RGC RF size. (A)** Representative time series of the eye position (X- and Y-coordinate of the pupil center extracted from eye-tracking camera images) during white-noise "checkerboard" stimulus presentation (black/green, centered and stable eye position used for reverse correlation analysis; gray, non-centered or non-stable period excluded from the analysis; probability distribution shown on the right), along with those of firing rate dynamics of two example RGCs and locomotion (from top to bottom), before (left) and after (right) ciproxifan administration. **(B)** Estimated spatial filter (receptive field) of the two example RGCs (top and middle) before (left) and after (right) ciproxifan administration. Note a shift of RF position of all recorded cells due to a shift of resting eye position after the treatment

(see panel A). **(C–F)** Cumulative distribution of the modulation index of RGCs before and after treatment (C, receptive field size; D, peak latency; E, peak frequency, F, mean firing rate): orange, chlorphenamine, $n = 6$ cells from 3 animals; green, ciproxifan, $n = 25$ cells from 4 animals; purple, PSAM/PSEM for HDC+ cells in non-TMN, $n = 9$ cells from 3 animals; red, PSAM/PSEM for HDC+ cells in TMN, $n = 8$ cells from 4 animals. Post-hoc test against saline control ($n = 17$ from 3 animals) after Kruskal–Wallis test: * $p < 0.05$; ** $p < 0.01$. Data and code underlying this figure are available at https://doi.org/10.5281/zenodo.17016431.
(TIF)

**S9 Fig. Histamine did not affect direction- or orientation-selectivity of RGCs or LGN cells. (A)** The average firing rate of a representative direction-selective (DS) RGC in response to moving gratings in eight different directions before (gray) and after treatment (color-coded): from left to right, PSAM/PSEM for TMN HDC+ cells (red) or non-TMN cells (purple), ciproxifan (green), chlorphenamine (orange), and saline (blue). **(B)** DS indices of RGC populations before and after treatment. Those with DS index $> 0.15$ in either condition were highlighted in the corresponding color (from left to right, $n = 11, 16, 31, 14$, and 30 RGCs, respectively). No significant change was observed in the DS index values: $p > 0.2$ in all cases (Wilcoxon signed-rank test). **(C)** Corresponding data for the OS indices of RGCs. No significant change was observed ($p > 0.7$ in all cases). **(D,E)** Corresponding population data for the DS and OS indices of LGN cells (from left to right, $n = 42, 76, 27, 58$ LGN cells; $p > 0.06$ and 0.3 in all cases, respectively). Data and code underlying this figure are available at https://doi.org/10.5281/zenodo.17016431.
(TIF)

**S10 Fig. Histaminergic effects on pupil dynamics are irrelevant to those on LGN visual responses. (A–C)** Comparison of the LGN population response properties between constricted and dilated pupil periods ($n = 350$ RGCs from 13 animals): A, peak latency, $51 \pm 20$ ms versus $48 \pm 19$ ms, median $\pm$ median absolute deviation, $p < 0.001$, Wilcoxon signed-rank test; B, peak frequency, $7.9 \pm 2.1$ Hz versus $8.1 \pm 2.4$ Hz, $p = 0.004$; C, mean firing rate, $11 \pm 7$ versus $15 \pm 7$ Hz, $p < 0.001$. **(D–G)** Comparison of the LGN population response properties between constricting and dilating pupil periods: D, ON-OFF polarity index of the temporal filters, $0.07 \pm 0.22$ versus $-0.06 \pm 0.27$, $p < 0.001$; E, peak latency, $49 \pm 19$ ms versus $49 \pm 19$ ms, $p = 0.8$; F, peak frequency, $7.9 \pm 2.0$ Hz versus $8.1 \pm 2.1$ Hz, $p = 0.3$; G, mean firing rate, $11 \pm 7$ Hz versus $14 \pm 7$ Hz, $p = 0.004$. **(H)** Comparison of peak latencies before treatment with constricted pupil versus after treatment with dilated pupil: from left to right, PSAM/PSEM for TMN HDC+ cells (red), ciproxifan (green), chlorphenamine (orange), and saline (blue): ** $p < 0.01$; *** $p < 0.001$, Wilcoxon signed rank test. **(I)** Comparison of peak latencies before treatment with dilated pupil versus after treatment with constricted pupil. Data and code underlying this figure are available at https://doi.org/10.5281/zenodo.17016431.
(TIF)

## Acknowledgments

The EMBL Light Imaging Facility is acknowledged for support in histological image acquisition and analyses; EMBL Gene Editing and Virus Facility for virus production; EMBL IT Support for provision of computer and data storage servers; and the LAR facility for taking care of animals. We thank Dmitry Molotkov, Tom Boissonnet, Vanesa Pelcastre, Sahana Rao, Shabnam Chandel, Lily Knowles for their help in experiments, and all the Asari lab members for many useful discussions.

## Author contributions

**Conceptualization:** Matteo Tripodi, Hiroki Asari.

**Data curation:** Matteo Tripodi.

**Formal analysis:** Matteo Tripodi, Hiroki Asari.

**Funding acquisition:** Hiroki Asari.

**Investigation:** Matteo Tripodi.

**Supervision:** Hiroki Asari.

**Visualization:** Matteo Tripodi, Hiroki Asari.

**Writing – original draft:** Matteo Tripodi, Hiroki Asari.

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
