## [Editor Report · Decision Letter 0]

18 Sep 2024

Dear Dr Asari,

Thank you for submitting your manuscript entitled "Histamine interferes with the early visual processing in mice" for consideration as a Research Article by PLOS Biology.

Your manuscript has now been evaluated by the PLOS Biology editorial staff as well as by an academic editor with relevant expertise and I am writing to let you know that we would like to send your submission out for external peer review.

Once your full submission is complete, your paper will undergo a series of checks in preparation for peer review. After your manuscript has passed the checks it will be sent out for review. To provide the metadata for your submission, please Login to Editorial Manager (https://www.editorialmanager.com/pbiology) within two working days, i.e. by Sep 20 2024 11:59PM.

Kind regards,

Christian

Christian Schnell, PhD

Senior Editor

PLOS Biology

cschnell@plos.org

---

## [Decision Letter · Decision Letter 1]

13 Nov 2024

Dear Dr Asari,

Thank you for your patience while your manuscript "Histamine interferes with the early visual processing in mice" was peer-reviewed at PLOS Biology. Your manuscript has been evaluated by the PLOS Biology editors, an Academic Editor with relevant expertise, and by several independent reviewers.

As you will see in the reviewer reports, which can be found at the end of this email, although the reviewers find the work potentially interesting, they have also raised a substantial number of important concerns. Based on their specific comments and following discussion with the Academic Editor, it is clear that a substantial amount of work would be required to meet the criteria for publication in PLOS Biology. However, given our and the reviewer interest in your study, we would be open to inviting a comprehensive revision of the study that thoroughly addresses all the reviewers' comments. Given the extent of revision that would be needed, we cannot make a decision about publication until we have seen the revised manuscript and your response to the reviewers' comments. Your revised manuscript would need to be seen by the reviewers again, but please note that we would not engage them unless their main concerns have been addressed.

We appreciate that these requests represent a great deal of extra work, and we are willing to relax our standard revision time to allow you 6 months to revise your study. Please email us (plosbiology@plos.org) if you have any questions or concerns, or envision needing a (short) extension.

**IMPORTANT - SUBMITTING YOUR REVISION**

*Resubmission Checklist*

*Published Peer Review*

*PLOS Data Policy*

*Blot and Gel Data Policy*

Sincerely,

Christian

Christian Schnell, PhD

Senior Editor

PLOS Biology

cschnell@plos.org

REVIEWS:

Reviewer #1: Review

Histamine interferes with early visual processing in mice.

In this manuscript, Tripodi and Asari investigate the effects of histamine on early visual processing in mice. They do so by combining (i) awake silicon probe recordings placed in the dLGN or optic tract, (ii) visual stimuli, (iii) a combination of pharmacological interventions to activate or suppress histaminergic receptors, (iv) pharmacogenetic perturbations (PSAM/PSEM), and (v) modeling. Their results and conclusions suggest that histamine release from the hypothalamic tuberomammillary nucleus modulates the rate and response kinetics of visual stimuli. These adaptations may have implications for state-dependent visuomotor processing.

The paper is well-written and presented, and the results are convincing. I have only a few major and minor points to discuss, which I believe need to be address prior to publication.

Major:

My main concern is behavioral control, which as the authors mention, could account for the observed effects. Thus, I would kindly ask for a more detailed behavioral analysis, not just the average but the distributions across animals. Let me be a bit more specific:

One of the main arguments goes: When applying H1 antagonist (mimicking a decrease of histamine), the pupil is dilated, using an H3 antagonist (mimicking an increase of histamine), the pupil is constricted. Counter to these pharmacological results, increasing histamine release using a chemogenetic approach has the opposite effect, constricting the pupil. These contrary results are taken as proof that the changes in the temporal dynamics are not related to luminance (pupile size changes). However, to be truly convincing, it would be necessary to show the dynamics of constriction with respect to the time the stimulus was recorded and the distribution of pupil sizes. The authors only show the relative size changes, but it is well possible that due to the manipulations, these start at a different baseline, are changed for a short period, or show different constriction dynamics. Thus, showing distributions with the absolute pupil sizes during the recording period would be more informative and compare the sizes during the recordings. For example, the authors could subsample times of similar pupil size to estimate the tempora filters. Ideally, one would use atropine to dilate the pupil together with the pharmacological perturbations. That would completely eliminate any luminance change and the difference can be directly attributed to the modulation histamine (perhaps even enhancing them), but a more detailed analysis of the behavior might be sufficient.

It would also be relevant to take a look at the eye movements. Are the position of the eyes changes, do the authors see more saccades? One indication of this is the presented RF in Supplementary Fig. 3. While the size of the RF changes, it seems that the position might also change. Could that be accounted for by more or less eye movements?

Regarding the locomotion speeds, the authors also use a very low threshold for "running" >2 cm/s. Moreover, only 20% of the time animals were "running". Taking the mean of many low entries is necessarily very low, this the ratio of treated and untreated, as plotted in Fig. 3B, C, might not be the best way to presen the results. I suggest that the authors should compare the distributions of absolute running speeds, and make sure that the periods used to characterize the response properties have similar behavioral properties (excluding running periods from the analysis, e.g.).

While the temporal filters are compelling, this is an indirect description of visual responses. Would the authors have examples of the same cell to repetitions of the same stimulus (e.g., full-field flashes) with different pharmacological perturbations? Perhaps the authors could find instances during their WN stimulus of identical changes an present them as a reaster plot? That would provide a direct link between the gain (changes in rate) and kinetics (speeds of the first spike) of the responses.

The statistical quantification in Figs. 1 and 2 are for the entire population. It would be good to compare all cells recorded for each animal. I would appreciate it if the authors could plot the distributions for all cells recorded for each animal as violin plots with each cell represented.

Finally, I'm curious about histaminergic projections to the retina. These projections have been reported recently and cited by the authors. I have heard, however, that the histaminergic projections to the retina are often not visible. I wonder if the authors see these projections in retinal explants. Either result is interesting since there are clear physiological effects. I wonder if they don't see them, if histamine could modulate retinal ganglion cell axons in the brain by changing the propagation of action potentials or synaptic release. I would be interested to know what the authors think.

Minor:

Title:

- I suggest changing the title to be more specific. "Interferes" is correct, but does not convey the essence of the finding. Knowing that gain and kinetics are modulated would attract more readers.

Abstract:

- Second sentence. Please change "even at the retina" to "already".

- "Our computational modeling analysis instead suggests neural circuit effects." Given that the previous sentence is about behavior, I would suggest for readability that you clearly define what the modeling suggests. Something like: "Instead, our computational modeling analysis suggests that the temporal response modulations are due to changes in the intrinsic properties of the circuit."

Main text:

Line 229. The sentence "we cannot deny" seems strange. I would suggest changing it to something of the sort: "Thus, it is possible tat the observed hitaminergic effects on the visual response properties in the muse early visual system are a result of correlated behaviorla modulation."

Reviewer #2: Based on in vivo electrophysiological recordings of visual responses in the optic tract and dLGN of awake head-fixed mice, both with and without pharmacological and chemogenetic manipulations, Tripodi et al. concluded that the early visual system, including retinal ganglion cells (RGCs) and dLGN neurons, exhibited reduced and slowed visual responses when histaminergic neurons in the tuberomammillary nucleus (TMN) of the posterior hypothalamus were activated. Furthermore, the effects of histaminergic modulation could not be attributed to changes in pupil dynamics or locomotion behavior. Instead, their computational modeling suggests that histaminergic modulation of the early visual system occurs via histamine H1 receptors in the retina. These findings are surprising, as high histamine levels are typically associated with high vigilance, yet here, high histamine resulted in weaker and slower responses in the early visual system, contrary to expectations.

Histaminergic modulation of early visual processing could be a significant topic in neuroscience. However, I have several major concerns about the experimental setup and results that need to be addressed to support and validate the authors' conclusions.

Major comments:

1. This manuscript was based on electrophysiology recordings but did not describe the spike sorting method used, nor whether single-unit signals were reliably isolated from their recordings.

2. It is unclear whether the comparison before and after chemogenetic or pharmacological manipulation was performed from the same single units. If so, evidence should be provided to demonstrate that the recordings from the same single units (especially from the optic tract) were stable over time.

3. The methods section did not specify where the probe was implanted in the optic tract or provide images of the implanted area. This lack of detail makes it impossible to rule out the possibility that the recorded signals from the optic tract originate from axons of neuron types other than retinal ganglion cells. For example, axons from the parabigeminal nucleus (PBG) follow the optic tract to project to the contralateral dLGN ('PBG axons coursed within the ipsilateral SOX toward the optic chiasm, crossing the ventral midline, and running dorsally within the optic tract en route to dLGN', Sokhadze et al., J Comp Neurol. 2021; see also Figure 1 in Tokuoka et al., J Neurophysiol 2020). Moreover, not all the retinal axons in the optic tract project to the dLGN.

4. This study suggests that the effects of chemogenetic activation of histaminergic neurons in the TMN on the early visual system are 'direct,' mediated by histamine receptors in the retina, rather than indirectly through modulating arousal or locomotion states. However, no evidence is provided to demonstrate that their TMN injection labeled any histaminergic axon terminals in the retina. This is especially concerning given that the number of labeled TMN neurons appears to be small (Figure S1), and histaminergic axons in the retina are known to be very sparse according to the literature.

5. The sample size is too small, with N = 2 to 3 under each condition. More mice should be recorded under each condition to strengthen the conclusions.

6. The concentrations of chlorphenamine (5 mg/kg) and ciproxifan (12 mg/kg) used are higher than those recommended in the literature, which could lead to side effects, including nonspecific binding of other receptors. For example, large doses of chlorphenamine may cause unwanted sedation, psychomotor retardation, and blockade of cholinergic and α-adrenergic receptors. For children, the dose is calculated based on age or body weight, ranging from 0.25 mg/kg for infants to 0.20 mg/kg for older children (https://www.drugs.com/uk/chlorphenamine-10mg-ml-solution-for-injection-leaflet.html). The ciproxifan dose used was also higher than usual: doi.org/10.1523/JNEUROSCI.22-16-07272.2002 used 1.5mg/kg; https://doi.org/10.1016/j.nlm.2010.10.008 used 3mg/kg).

7. Brain histamine level has a circadian pattern. The manuscript should specify the time of day when recordings for each condition were performed. Were the control and experimental condition recordings consistently performed around the same period of the day?

8. It is surprising that applying an H1 antagonist led to larger pupil sizes and a substantial decrease in no-running periods, opposite to what would be expected since histamine is usually associated with high vigilance.

Minor comment:

Please cite references for the statement: "In particular, many prey species often sleep with their eyes open, including mice."

Reviewer #3: In the current study, Tripodi and Asari investigate how the neuromodulator histamine affects early visual processing in mice. Using pharmacological and chemogenetic tools in awake head-fixed mice, the authors found that histaminergic projections from the tuberomammillary nucleus (TMN) of the posterior hypothalamus slowed down and weaken the visual responses of retinal ganglion cells (RGCs) and cells in the lateral geniculate nucleus (LGN). After evaluating the effects of their pharmacological interventions on pupil dynamics and locomotion (running activity) and finding no clear correlations, the authors argue that the observed histamine effects likely arise from gain modulation via H1 receptors in the retina. In addition, the authors use simple computational modeling to investigate the underlying mechanism. The authors speculate that histaminergic modulation in the early visual system may be ethologically beneficial for nocturnal animals, such as mice, as this could allow for faster responses to visual threats when histamine levels are low (i.e. during less active periods).

Strengths:

This work is experimentally strong: The authors combine electrophysiological in vivo recordings with pharmacological perturbations of RGC and LGN activity, chemogenetic manipulation of histaminergic cells in the TMN, behavior as well as computational analyses to dissect the effects histamine on visual signal processing starting in the retina. In particular, the comparison between the pharmacological and chemogenetic approaches is instructive as it shows that similar manipulations of the same pathway can lead to different effects (i.e., if the release neurons are directly manipulated or if the presynaptic autoreceptors are modulated).

Major points:

Unfortunately, in its current form, the manuscript falls short in several aspects.

(1) The authors collected a very interesting and relevant dataset but the discussion of the results is weak. It almost seems as if the findings that histamine (which is expected to be high in periods of activity / during day time) slows and reduces activity in the early visual system surprised the authors, and they did not know how to reconcile this with expectations and earlier studies. As a result, the discussion is rather superficial. For instance, it remains unclear why mice need faster responses when they are asleep but not during their activity periods, in particular as predation seems likely whenever mice are active (and visible). How does this relate to findings in other species? The authors mention experiments in rabbits ("histamine leading to slow depolarization in LGN cells") but do not put this in context with their findings in mice.

(2) A recent paper by Warwick et al. (2024) studied histaminergic feedback in the early visual system in detail and partially with similar methods like the authors. Tripodi and Asari cite the paper but fail to discuss their results in the context of the findings by Warwick et al. In view of the overlap between the two studies, it is unclear to me why this opportunity of discussing both sets of findings in a broader context was missed. Are the results consistent or where do they diverge? For instance, Warwick et al. reported specific histaminergic effects on direction-selective ganglion cells; have the authors looked into this aspect?

(3) To better appreciate the results, it is needed to discuss histamine release and experimental conditions in more detail: Is histamine released more dependent on sleep/activity or does the day-night cycle also play a role? In this context, when were the experiments performed? Always at the same time of the day? When relative to the day-night cycle of the animals? Could the fact that the H1 antagonist had smaller effects on neural activity be explained by a low endogenous level of histamine during the experiments?

(4) The conclusion that histamine action in the retina is the main driver for the observed effects largely rests on the behavioral data. Can the authors provide supporting statistics for the pupil dynamics/locomotion data in support of their claim?

(5) It is unclear what the model adds: Specifically, it is difficult to relate the data (Fig. 4F-I) to the model predictions (Fig. 4A-E). Regarding the data: Cell types and experimental conditions are resolved in the scatter plot but then all data are fit together? The logic behind this needs to be explained.

Minor points:

Why are the authors not showing a comparison of the data before/after saline injection (i.e. in Figs. 1 and 2)?

In Suppl. Fig. 3C, all effects go in the same direction, suggesting that RF sizes shrink independent of the manipulation. In Suppl. Fig. 3E, PSAM and the H3 antagonist act differently. Please discuss.

How effective is the chemogenetic activation of HDC+ cells in TMN, that is, how many histaminergic neurons in the TMN are likely to be activated? In this context, what are the endogenous levels of histamine released from the TMN to the retina?

What about the co-release of GABA and its potential other interactions mentioned by the authors? Could GABA effects explain the difference between the chemogenetic and H1 antagonist perturbation?

Why did the authors manipulate H1 and H3 receptors and not H2? Please explain.

The functional classification of RGC and LGN types using temporal STAs obtained from responses to full-field white-noise stimuli is limited. The authors classified the responses into six broad functional groups, separated by response polarity and kinetics. However, there are other cell types found in the LGN, such suppressed-by-contrast cells, which they cannot identify using this approach. In any case, the authors perform this classification but do not make use of it. Were there any cell type-specific histaminergic effects?

How were the pharmacological agents applied? Systemically? What was the rationale for the selection of the concentrations of the H1 and H3 agonists/antagonists? Were the observed effects reversible?

Methods: Use true typesetting for equations

Methods: Recordings in dLGN and from optic nerve are not described (only surgery). Was the same surgery used for both recordings and viral injections?

line 48: "Such retinal mechanisms of neuromodulation are, however, much slower than needed to modulate retinal function to follow rapid behavioral changes." This is a bold statement, are there any references supporting this?

Fig 1 & 2: Suggestions: Introduce symbols for manipulations already in (C) to (E). in 1G,H, show respective "rest" of synapse with gray outline, then the synapse and the site of action is easier to understand

Suppl. Fig. 2: Indicate cell types in (A) and (C) also in color instead of only symbols.

---

## [Decision Letter · Decision Letter 2]

26 Aug 2025

Dear Dr Asari,

Thank you for your patience while we considered your revised manuscript "Central histaminergic system reduces response gain and slows visual processing in the retina and lateral geniculate nucleus of awake mice" for publication as a Research Article at PLOS Biology. This revised version of your manuscript has been evaluated by the PLOS Biology editors, the Academic Editor and two of the original reviewers.

Based on the reviews and on our Academic Editor's assessment of your revision, we are likely to accept this manuscript for publication, provided you satisfactorily address the remaining points raised by the reviewers. Please also make sure to address the following data and other policy-related requests:

* We would like to suggest a different title to improve its accessibility for our broad audience:

The central histaminergic system slows visual processing in the retina and lateral geniculate nucleus of awake mice

* DATA POLICY:

Regardless of the method selected, please ensure that you provide the individual numerical values that underlie the summary data displayed in the following figure panels as they are essential for readers to assess your analysis and to reproduce it: 4LM, S3BC, S4CDEF, S5AB, S7CDEF and S10HI.

* CODE POLICY

We expect to receive your revised manuscript within two weeks.

*Published Peer Review History*

*Press*

Sincerely,

Christian

Christian Schnell, PhD

Senior Editor

cschnell@plos.org

PLOS Biology

Reviewer remarks:

Reviewer #1: I’m happy with the current form. There is just one little misspelling: I believe "specific-specific" should be "species-specific" in line 545.

Reviewer #3 (Dominic Gonschorek, Thomas Euler have signed the report): The authors did a great job in answering our questions and revising the manuscript.

We have no further comments.

---

## [Editor Report · Decision Letter 3]

9 Sep 2025

Dear Dr Asari,

Thank you for the submission of your revised Research Article "The central histaminergic system slows visual processing in the retina and lateral geniculate nucleus of awake mice" for publication in PLOS Biology. On behalf of my colleagues and the Academic Editor, Tom Baden, I am pleased to say that we can in principle accept your manuscript for publication, provided you address any remaining formatting and reporting issues. These will be detailed in an email you should receive within 2-3 business days from our colleagues in the journal operations team; no action is required from you until then. Please note that we will not be able to formally accept your manuscript and schedule it for publication until you have completed any requested changes.

PRESS

Sincerely, 

Christian

Christian Schnell, PhD

Senior Editor

PLOS Biology

cschnell@plos.org